# Parametric Design Method and Lift/Drag Characteristics Analysis for a Wide-Range, Wing-Morphing Glide Vehicle



**Zikang Jin** [1], **Zonghan Yu** [1,*], **Fanshuo Meng** [1], **Wei Zhang** [2], **Jingzhi Cui** [2], **Xiaolong He** [3], **Yuedi Lei** [4] **and Omer Musa** [4]

1    School of Mechanical and Materials Engineering, North China University of Technology, Beijing 100144, China; 2021312070121@mail.ncut.edu.cn (Z.J.); 15373455942@163.com (F.M.)
2    Beijing Aerospace Systems Engineering Research Institute, Beijing 100076, China; zw_xidian@163.com (W.Z.); 13521904775@139.com (J.C.)
3    China Academy of Launch Vehicle Technology, Beijing 100076, China
4    College of Energy and Power, Nanjing University of Aeronautics and Astronautics, Nanjing 210016, China; lyd6096@nuaa.edu.cn (Y.L.); omermusa@nuaa.edu.cn (O.M.)
*    Correspondence: yzh@ncut.edu.cn

**Abstract:** The parametric design method is widely utilized in the preliminary design stage for hypersonic vehicles; it ensures the fast iteration of configuration, generation, and optimization. This study proposes a novel parametric method for a wide-range, wing-morphing glide vehicle. The whole configuration, including a waverider fuselage, a rotating wing, a blunt leading edge, rudders, etc., can be easily described using 27 key parameters. In contrast to the typical parametric method, the new method takes internal payloads into consideration during the shape optimization process. That is, the vehicle configuration can be flexibly adjusted depending on the internal payloads; these payloads may be of random amounts and have different shapes. The code for the new parametric design method is developed using the secondary development tools of UG (UG 10.0) commercial software. The lift and drag characteristics over a wide operational range (H = 6–25 km, M = 2.5–8.5, AOA = 0–10°) were numerically investigated, as was the influence of the retracting angle of the morphing wings. It was found that, for the mode of the fully deployed wings, the lift-to-drag ratio (L/D) remained at a high level ($\geq$4.7) over a Mach range of 4.0–8.5 and an AOA range of 4–7°. For the mode of the fully retracted wings, the drag coefficient remained smaller than 0.02 over a Mach range of 4.0–8.5 and an AOA range of 0–5°. A wide L/D of 0.3–4.7 could be achieved by controlling the retracting angle of the wings, thus demonstrating a good potential for flight maneuverability. The flexible change in L/D proved to be a combined result of varying pressure distribution and edge-flow spillage. This will aid in the further optimization of lift/drag characteristics.

**Keywords:** hypersonic; wing morphing; wide range; UG secondary development; parameter modeling; aerodynamic shape optimization; numerical simulation

## 1. Introduction

With the advancement of aerospace technology, the performance requirements for hypersonic vehicles are no longer defined in terms of a specific Mach number. Achieving a large airspace and a wide speed range has become a new goal in aerodynamic shape design [1]. The advantage of morphing technology lies in its ability to adaptively alter the shape based on changes in the flight environment and mission requirements, thereby accommodating various flight conditions and expanding the application range [2]. However, the current research on morphing, such as that on overall morphing schemes, deformation mechanisms, structural transformations, and aerodynamic shape optimization, primarily focuses on the low speed to the supersonic range, with a notable lack of study on the hypersonic range. Therefore, combining hypersonic vehicles and variant technology to achieve high-speed flight with a wide range is a research direction worth exploring.

For both morphing and fixed-configuration vehicles, aerodynamic shape optimization is an indispensable part of the overall design and evaluation. It is typically based on the parametrization of the vehicle. The aim of parametrization is to minimize the number of parameters used while fully capturing the shape of the vehicle, thereby enabling the achievement of optimal configurations at reduced computational costs during the process of shape optimization. Lift/drag characteristics, handling and stability characteristics, and other indicators are important criteria for judging the performance of an aerodynamic shape. Additionally, heat protection, internal volume, and other practical requirements also play a crucial role in engineering applications. Xu and Yuan [3,4] used the three-dimensional class and shape transformation (CST) method to parameterize the vehicle model in their respective studies. The former used volume ratio, lift-to-drag ratio, and heat flux density as indicators to conduct multi-objective vehicle optimization; the latter considered shape, size, volume, area, and aerodynamic heat to optimize the vehicle. Hao [5] used a piecewise, third-order B-spline and other curves to parameterize the model of the three-dimensional section of the vehicle, and then optimized the upper surface by changing the upper wall profile, with volume and the lift-to-drag ratio as the two objectives. Ma [6] established a multidisciplinary model like that of the X-51A configuration and carried out collaborative optimization with the range as the goal, reducing a small quality in exchange for a large range increase. Viviani [7] employed the free-form deformation (FFD) method for the multidisciplinary design of re-entry vehicles, focusing on aerodynamic performance and thermal protection as the optimization objectives under typical conditions along the re-entry trajectory. Brian [8] applied multidisciplinary optimization using machine learning, targeting flight performance in multiple flow conditions along the flight trajectory. Liu and Peng [9,10] used the aerodynamic layout of the waverider configuration and used the generation parameters of the cone-guided waverider as variables to optimize the L/D and volume ratio but did not set the blunt leading edge. Ma [11] used the CST method for a lifting body configuration, optimizing the L/D and volume ratio by using a multi-objective evolutionary algorithm based on the decomposition (MOEA/D) method. Liu [12] used genetic algorithms to optimize the L/D for waverider configurations. Most of the shape optimization studies of vehicles use volume, volume ratio, or mass as optimization indicators. These optimization studies use the above indicators as their constraints; then, they calculate the aerodynamic performance of the vehicles under different independent variables to find the optimal L/D configuration. However, in practical engineering applications, it is necessary to consider the dimensions of various internal payloads in the design of the shape.

Lobbia [13–15] constructed a waverider configuration based on a B-spline, combining L/D, volume ratio, and a cylindrical internal load with well-defined dimensions; the aim was a multi-objective optimization of the aerodynamic shape, and it was validated using wind tunnel testing. Peng [16] used a clamped third-order spline method for vehicle parametrization and used a two-dimensional approach to represent a cylindrical internal payload; this resulted in a strong coupling between the vehicle shape and the internal payload shape. This method was specifically used to optimize the L/D and was compared with traditional methods that use volume ratio and L/D as optimization goals. Zhang [17] used a non-uniform rational B-spline (NURBS) and FFD technique in the parameterization of the lifting body configuration. According to the study, this method showed greater deformation capabilities than the other FFD methods. In subsequent shape optimization efforts, the optimization was carried out by focusing on a conical internal payload and L/D. In summary, compared with the optimization of the overall plot ratio, the specific shape of the internal payload is considered within the scope of the shape optimization target; this can better meet the needs of practical engineering applications. However, this area of research is still relatively underexplored, and optimized methods for cylindrical or conical payload shapes often fall short when applied to internal payloads of arbitrary shapes.

Yao [18] undertook secondary development based on CATIA (CATIA V5R14) software, optimizing the arrangement of internal payloads to maximize the utilization of the

internal space. To prevent interference between internal payloads and the fuselage during the optimization process, the interference-checking function of CATIA played a crucial role. Although his work did not involve the optimization of the external shape, the methodology employed is significantly relevant to this study.

This study proposes a parametric design approach for hypersonic wing-morphing glide vehicles, followed by the creation of a corresponding modeling program based on UG secondary development (Unigraphics NX, (Plano, TX, USA)), along with a matching interactive interface. Through this interface, users can input parameters to achieve whole-vehicle modeling in a short period. In terms of shape optimization, the interference-checking function in UG is integrated into the algorithm, allowing automatic adjustments to the shape to accommodate internal payloads of any shape and quantity, thereby enhancing design efficiency. Furthermore, with subsequent manufacturing processes in mind, the optimization method is further refined by combining the bisection method with the distance measurement functions within UG into the optimization program. This combination enables precise control of the minimum distance between the shape and the internal payloads and has significant value for engineering applications.

## 2. Design and Optimization of the Vehicle Shape

### 2.1. Aerodynamic Configuration

The aerodynamic configuration of the vehicle utilizes a waverider configuration, as depicted in Figure 1, in which the fuselage is represented in blue. During high-speed flight, the fuselage can generate a waverider effect to a certain extent, creating a high-pressure zone under the vehicle and thus enhancing the lift-to-drag characteristics of the vehicle. The upper surface is characterized by being higher in the middle and lower on both sides, and it features a larger payload space in the central area. Moreover, compared to other designs, the pressure from the incoming airflow is primarily concentrated in the central raised area, with reduced drag on both sides. This configuration ensures that the overall drag on the upper surface is minimized when accommodating internal payloads.

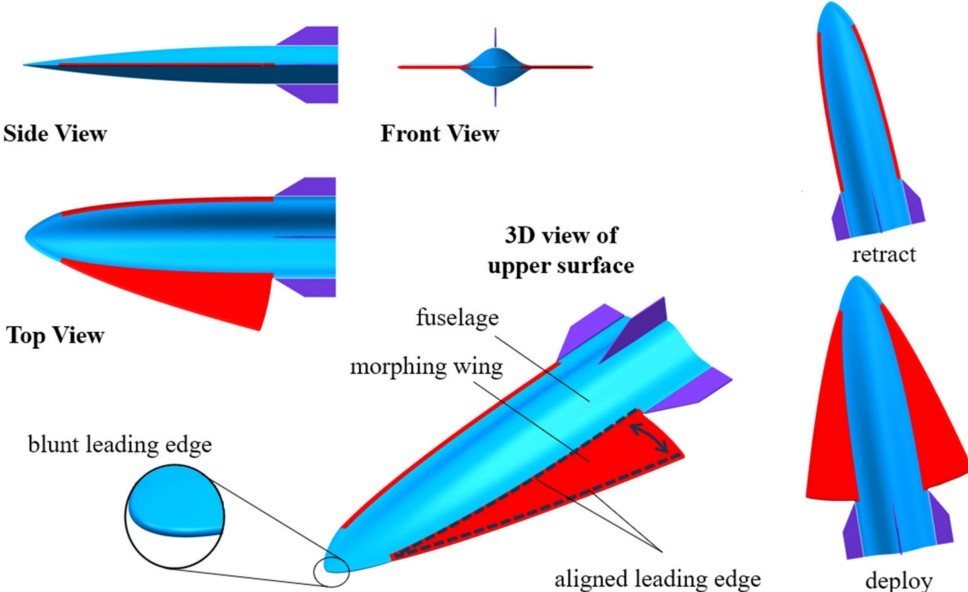

**Figure 1.** Shape of the vehicle.

Both leading edges of the wings and fuselage need to be blunted. If the wings were to utilize the traditional semicircular blunting method, the radius would be determined by the vertical distance at the blunting root, namely the thickness of the wings, preventing designers from altering the radius of the blunt leading edge as required by the design. Therefore, a blunt leading edge configuration is adopted, as shown in Figure 2; it combines

two transition lines (second-order Bessel curves) with a circular arc, where the blunting radius R can be freely chosen; T represents the thickness of the wing, parameter A determines the length of the blunt leading edge, and parameter B controls the shape of the transition curve. To maintain consistency between the wing-leading edge and the fuselage, the rear edge of the fuselage adopts the same sectional shape as the wing-leading edge.

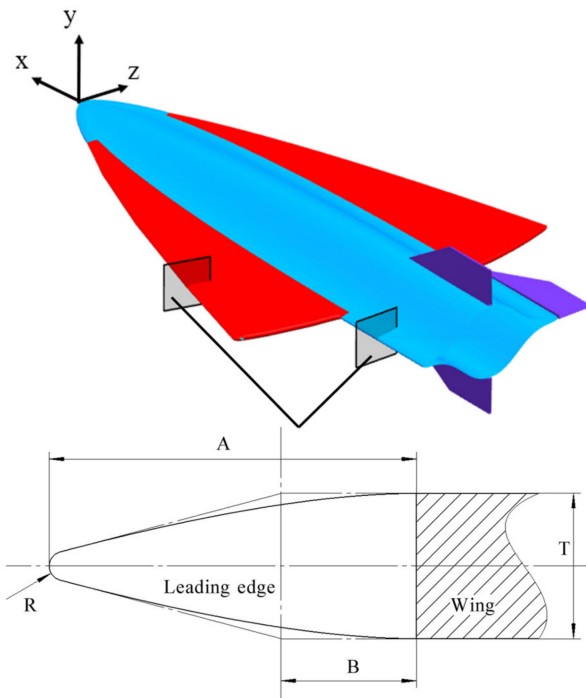

**Figure 2.** Normal section of leading edge on rear fuselage and wing.

*2.2. Parameterization Method of Vehicle*

The two-dimensional B-spline method [19] is used to parameterize the whole vehicle. The B-spline is widely used in parametric modeling and optimization due to its local modifiability and flexible control property [20–22]. The shape of the vehicle is cut on a three-dimensional section, and the B-spline is used as its cross-section line to express the surface characteristics. The general parametric equation for an n-order B-spline is expressed as [23]:

$$B(t) = \sum_{i=0}^{n} P_i C_i^n (1-t) t^i \tag{1}$$

As shown in Figure 3, this study combines second- or third-order clamped boundary B-splines with arcs and straight lines to express the sectional lines. The side view represents the upper and lower surfaces of the fuselage, with the sectional lines consisting of the blunt leading edges (arc), upper and lower surfaces (second-order clamped boundary B-splines), and rear section (a straight line). Taking the profile lines of the upper surface as an example, the equations for the arc and the straight line are readily determined. The second-order clamped boundary B-spline is fully constrained by three control points, and obtaining its parametric equation should begin with the determination of the tangent point to the arc, denoted as $(x_0, y_0)$:

$$x_0 = \frac{R[H_1^2 + (S_1 - L_1)^2 - R(S_1 - L_1)] + RH_1\sqrt{H_1^2 + (S_1 - L_1)^2 - 2R(S_1 - L_1)}}{R^2 + (S_1 - L_1)^2 + H_1^2 - 2R(S_1 - L_1)} \tag{2}$$

$$y_0 = \sqrt{-2Rx_0 - x_0^2} \tag{3}$$

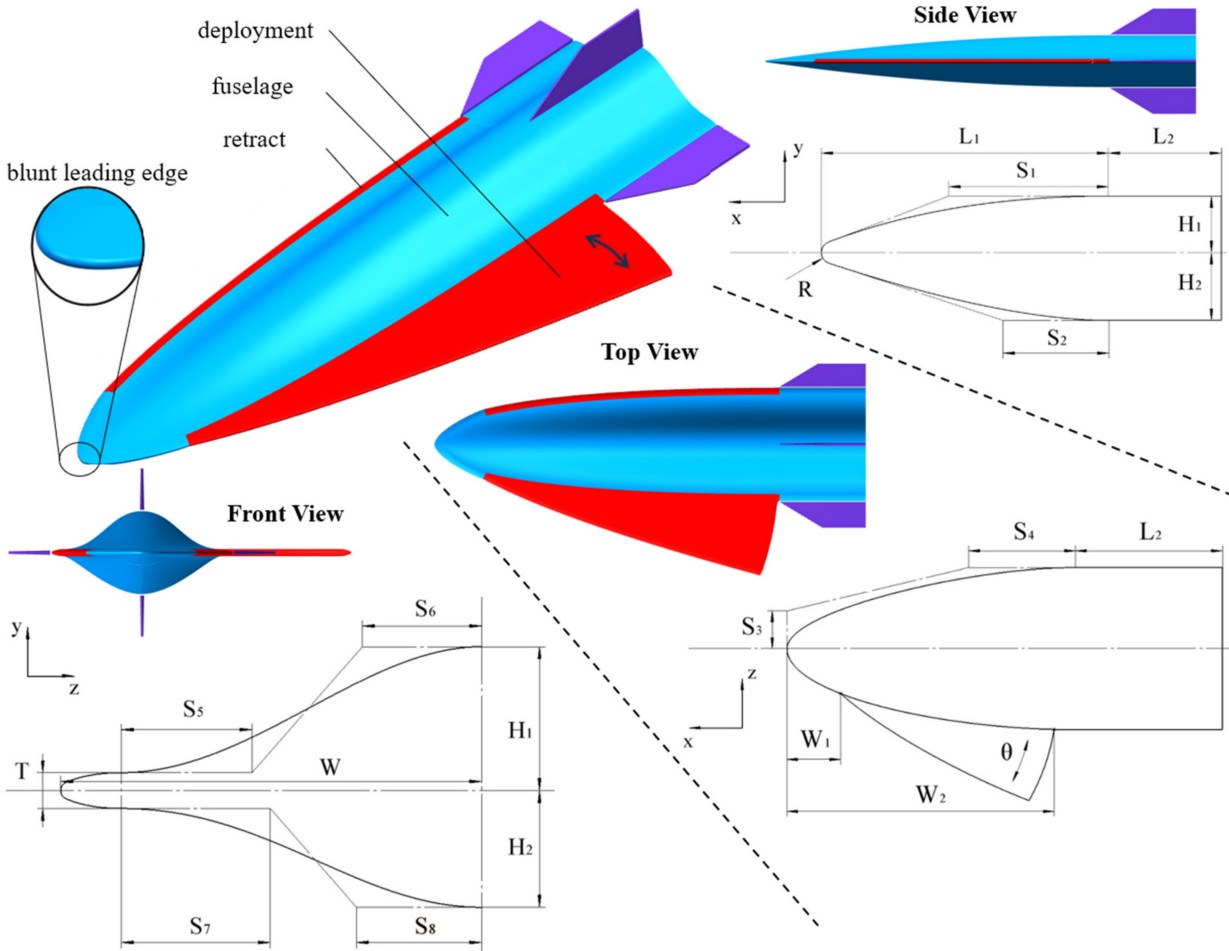

**Figure 3.** Three-dimensional cross-sections of the vehicle.

The uppercase variables are shown in Table 1, and the coordinates of the other two points are $(S_1 - L_1, H_1)$ and $(-L_1, H_1)$. By substituting them with $(x_0, y_0)$ in Equation (1), the curve parameter equation is obtained as:

$$\begin{cases} x(t) = (1 - t)^2 x_0 + 2t(1 - t)(S_1 - L_1)t^2 - L_1 \\ y(t) = (1 - t)^2 y_0 + 2t(1 - t)H_1 + t^2 H_1 \end{cases}, \; t \in [0, 1] \qquad (4)$$

**Table 1.** Governing parameters of current design.

| Serial | Part | Description/Name | Initial Value |
|--------|------|------------------|---------------|
| 1 | | Upper height/$H_1$ | 0.3 m |
| 2 | | Lower height/$H_2$ | 0.3 m |
| 3 | | Bend length/$L_1$ | 4 m |
| 4 | | Stright length/$L_2$ | 1 m |
| 5 | | Half width of the fuselage/$W$ | 0.6 m |
| 6 | | Pole 1 of B-spline/$S_1$ | 2 m |
| 7 | Fuselage | Pole 2 of B-spline/$S_2$ | 2 m |
| 8 | | Pole 3 of B-spline/$S_3$ | 0.5 m |
| 9 | | Pole 4 of B-spline/$S_4$ | 3 m |
| 10 | | Pole 5 of B-spline/$S_5$ | 0.275 m |
| 11 | | Pole 6 of B-spline/$S_6$ | 0.18 m |
| 12 | | Pole 7 of B-spline/$S_7$ | 0.055 m |
| 13 | | Pole 8 of B-spline/$S_8$ | 0.24 m |
| 14 | | Wing start point/$W_1$ | 0.6 m |
| 15 | Wing | Wing end point/$W_2$ | 3.98 m |
| 16 | | Wing angle/$\theta$ | 15° |
| 17 | | Wing thickness/$T$ | 0.05 m |

**Table 1.** *Cont.*

| Serial | Part | Description/Name | Initial Value |
|---|---|---|---|
| 18 | | Radius of rudder front/$R_r$ | 0.006 m |
| 19 | | Up width of rudder/$T_{r1}$ | 0.018 m |
| 20 | | Down width of rudder/$T_{r2}$ | 0.03 m |
| 21 | Rudder | Rudder length/$L_r$ | 1 m |
| 22 | | Rudder height/$H_r$ | 0.3 m |
| 23 | | Distance between horizon rudders and fuselage/$D_1$ | 0.01 m |
| 24 | | Distance between horizon rudders and fuselage/$D_2$ | 0.01 m |
| 25 | | Radius of leading edge/$R$ | 0.01 m |
| 26 | Leading edge | Length of blunt/$A$ | 0.075 m |
| 27 | | Control point of transition curve/$B$ | 0.0375 m |

As indicated by Equation (4), the profile line of the upper surface is fully constrained by the parameters $R$, $L_1$, $H_1$, and $S_1$, with the first three usually determined at the beginning of the design. Therefore, $S_1$ is chosen as the curve control parameter and incorporates $S_1$ into the subsequent optimization work.

In Figure 3, the top view and front view are primarily composed of third-order clamped boundary B-spline curves. The generation process is similar to that of the above; so, it is no longer described.

The air rudders employ a cross arrangement, and the single rudder model is shown in Figure 4.

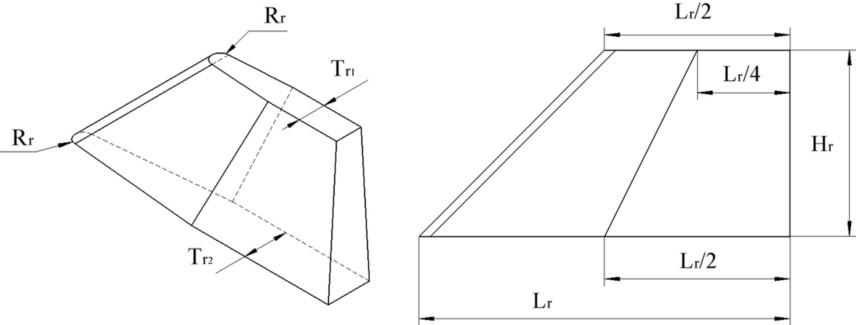

**Figure 4.** Diagram of single rudder.

The parameterized results, encompassing a total of 27 parameters, can fully represent the characteristics of the whole vehicle; the parameters are listed in Table 1.

Parametric modeling is a method in design and engineering that uses parameters to define and manipulate the characteristics of a model. It usually needs the designers to perform the modeling work by following specific steps to refine the dimensional constraints and to establish the corresponding parameters. This approach enables automatic adjustments in related sections of the model with a single parameter change during subsequent modifications, thereby enhancing design efficiency [24]. The traditional parametric method based on different CAD software (like Solidworks, CATIA) requires numerous manipulations on interactive interfaces and thus consumes more time. In contrast, parametric modeling technology based on UG secondary development allows programming and compilation through tools like UG/OPEN, VS, etc., eliminating the need for multiple manipulations on interactive interfaces. Users only need to input the required parameters, and the program automatically generates the whole model. This method makes model establishment and modification faster and facilitates the subsequent optimization steps; thus, it represents an excellent method for parametric modeling. It has been applied in the research of various model designs [25–28].

To further reflect the design requirements, before initiating the design modeling process, a user interface is established using the UI Styler module within UG, as shown in Figure 5. The users input parameters and press the 'confirm' button; the initial data are then transferred to the program through UG/Open API, concurrently initiating the modeling

program. The input parameters are listed in Table 1. The overall framework of the program is illustrated in Figure 6. It begins with data reading and determines whether to generate the wings. Once the criteria are met, it operates within UG to construct curves on various sections of the vehicle and establishes a comprehensive set of dimensional constraints and an expressions library. The advantage of the method lies in its power to aid users in modifying the model and laying the foundation for subsequent shape optimization. After all of the exterior curves are established, the program employs a series of surface modeling functions within UG to connect the curves, ultimately accomplishing the construction of the entire vehicle.

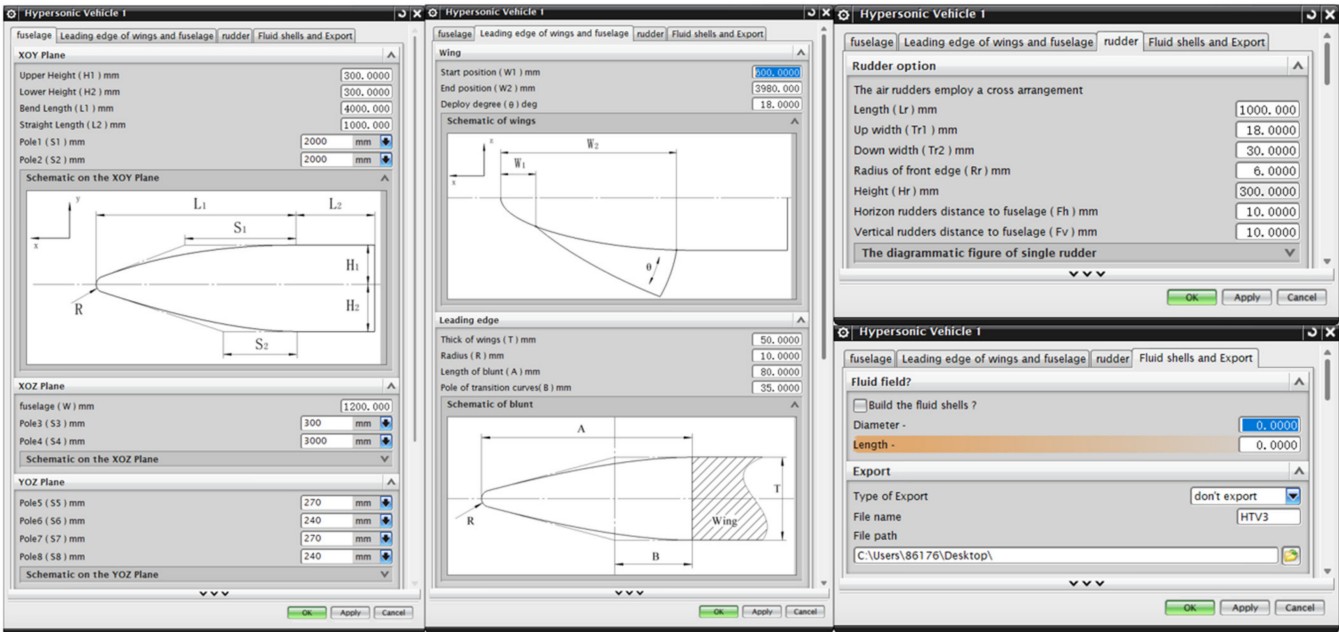

**Figure 5.** Interactive interface.

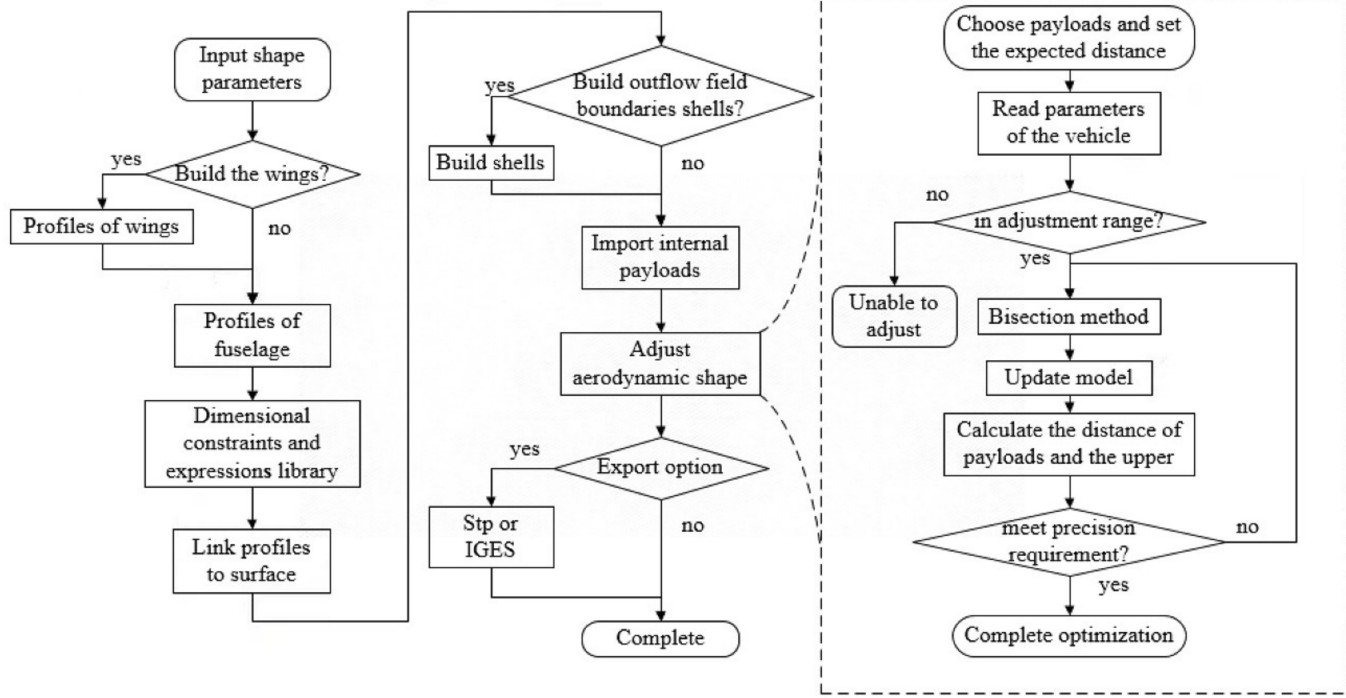

**Figure 6.** The program chart of parametric modeling and shape optimization.

In addition to its core functionalities, the program is enhanced with the capability to establish flow field boundaries and export the model. Users can input the desired diameter and length of the circular outflow field, and the program will construct the shell of the corresponding circular outflow field accordingly. Furthermore, there is an option to export the file in various formats, such as .stp and .iges. The exported model files can be directly used in software like ICEM CFD (2019 R3) for meshing, which facilitates the subsequent computational tasks for the user. All of the aforementioned processes can be completed in less than half a minute on a standard personal computer, significantly increasing the design efficiency of the vehicle.

*2.3. Shape Optimization*

Most hypersonic vehicles are flat and have the disadvantage of a low volume ratio [29]. Therefore, improving the volume ratio is generally one of the main optimization objectives of aerodynamic shape construction. Most of the studies use the following formula to represent the plot ratio ($\eta$) [11,30–32]:

$$\eta = \frac{6V(\pi)^{1/2}}{A_w^{3/2}} \tag{5}$$

where $V$ represents the volume, and $A_w$ denotes the wetted surface area of the vehicle. The volume ratio can, to a certain extent, indicate the payload capacity of the vehicle. However, in engineering applications, different departments often design the internal components separately, leading to diverse shapes in the design outcomes. Therefore, shape optimization based on the volume ratio generally cannot meet the loading requirements of given specific dimensions.

To address the aforementioned issue and better adapt the vehicle shape to internal payloads, this parametric-modeling-based study develops a self-adjustment program for the upper surface of the fuselage. This program can adjust the upper surface to any shape and quantity of the internal payloads, ensuring that all payloads can be housed within the fuselage. Furthermore, considering that margins are often left in the design process of each component, the program achieves precision control of the distance between the internal payloads and the upper surface by incorporating the bisection method with the distance judgment functions in UG, based on the interference checking.

The program flowchart, which is shown within the dashed box in Figure 6, primarily follows the steps of the bisection method, the updating of the model, and the calculation of the distance. After each parameter adjustment, model updating is required and is executed through the 'UF_MODL_update' function, while the distance from internal payloads to the upper surface can be calculated using the 'UF_MODL_ask_minimum_dist' function. If this meets the case of intersection, the distance is returned as 0 [33].

The parameter $S_1$ for the upper surface is taken as the independent variable, and the distances $D_i (i \in [1, n])$ from n positive internal payloads to the upper surface is taken as the optimization parameter, with the desired spacing d as the optimization goal. Initially, a mathematical model is established; the relationship between $S_1$ and $D_i$ is represented by the function set $F_i(S_1)(I \in [1, n])$, with $S_1 \in [0, L_1]$.

The program process of the optimization step can be summarized as follows:

**Pre step.** Set n, d, Let x = $L_1$.

**Step 1.** $D_i = F_i(S_1)$, $D_i = [D_0, D_1, \ldots, D_n]$, $F_i(S_1) = [F_0(S_1), F_1(S_1), \ldots, F_n(S_1)]$.

**Step 2.** Let $S_1 = 0$, if $\min(D_i) < d$: Unable to adjust and go to **End.**

**Step 3.** Let $S_1 = L_1$, if $\min(D_i) > d$: Unable to adjust and go to **End.**

**Step 4.** x = x/2, $S_1 = S_1 + x$, jump **Step 5.**

**Step 5.** x = x/2, $S_1 = S_1 - x$.

**Step 6.** $D_i = F_i(S_1)$.

**Step 7.** if abs($\min(D_i) - d$) < err: go to **End.**

**Step 8.** if $\min(D_i) > d$: go to **Step 4**; else: go to **Step 5.**

**End.**

In the process, 'err' is designated as the convergence accuracy, which is measured in meters, while 'x' acts as an intermediary variable.

A program was designed based on the optimization steps outlined above, and a corresponding interactive interface was also incorporated. As depicted in Figure 7c, when the user builds the whole shape of the vehicle and imports the internal loads (four distinct shapes of internal loads are illustrated as ①, ②, ③, ④ in Figure 7c), the 'Select Body' option within the interactive interface allows for the selection of all the internal loads. After inputting the desired spacing into the interface, the program automatically adjusts the upper surface to accommodate those internal loads with 'err' as the target accuracy (shown as $10^{-7}$ m in the Figure 7). The situation of the operational procedure is presented in Figure 7a,b, where Figure 7a reveals the changing distance $D_i$ ($i \in [1, 4]$) from the four types of internal loads to the upper surface and the variation in S1 with the times of the loop computing. Finally, 'payload 1' is the closest to the upper surface; Figure 7b exhibits the convergence of the difference between its distance to the upper surface and the desired distance in the loop computing; the difference typically converges to a precision of $10^{-7}$ m within just 20 steps of loop computing. The execution time of the optimization program does not exceed one minute, which significantly enhances the optimization efficiency. Upon completion of the optimization program, the optimization parameter $S_1$ converges to 2.339 m. Subsequently, the numerical simulation is performed based on the optimized shape.

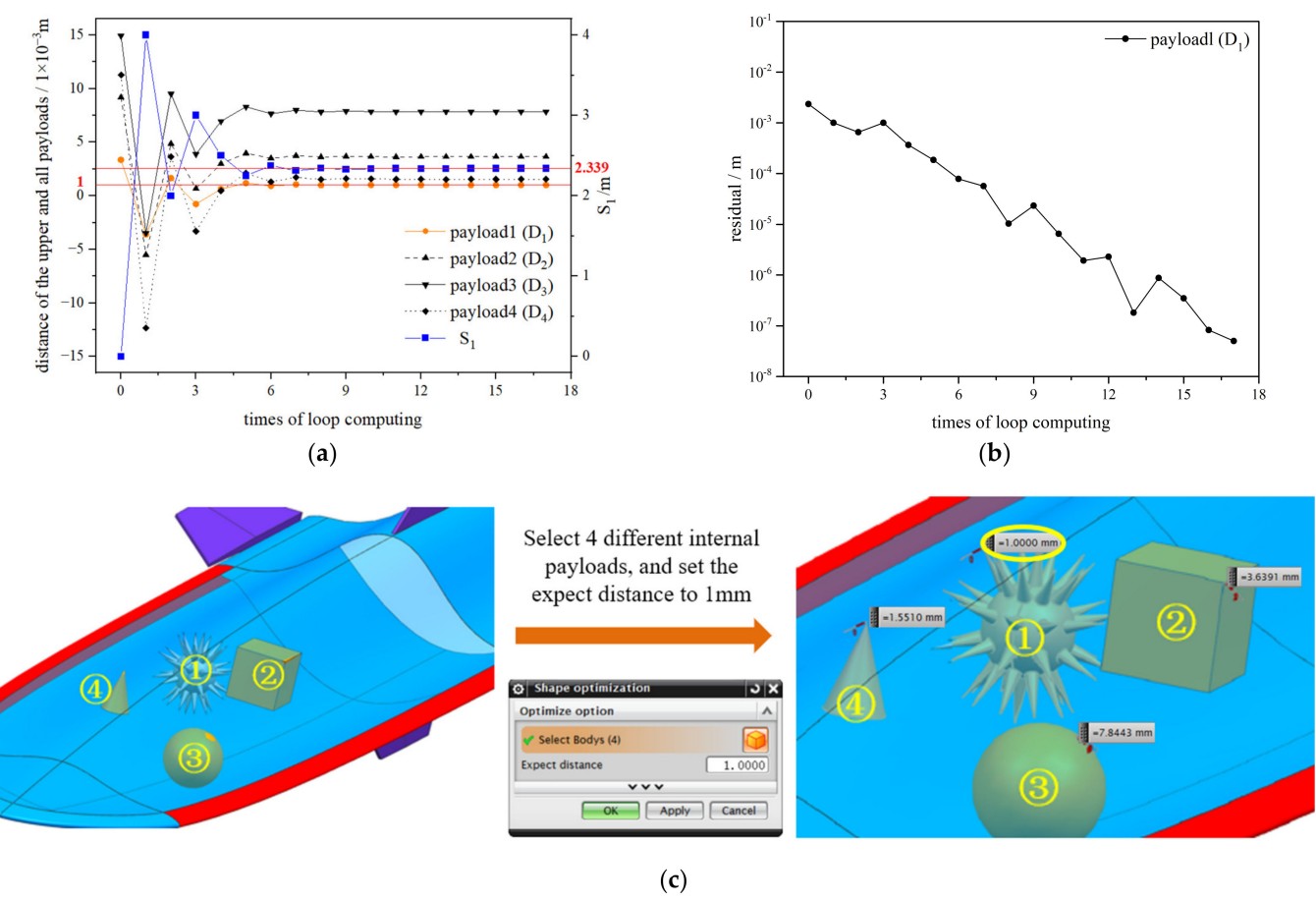

**Figure 7.** Optimization process for four arbitrary shapes of internal payloads: (**a**) the variation of $S_1$, $D_{1-4}$ with times of loop computing; (**b**) the variation in residual with times of loop computing; (**c**) process of optimizing.

## 3. Numerical Simulation

### 3.1. Settings

The mesh generation and numerical simulation are conducted using ANSYS (2019 R3) software (Canonsburg, PA, USA). The grid is divided using ICEM CFD, and Figure 8 shows the grid of the vehicle with the wings deployed.

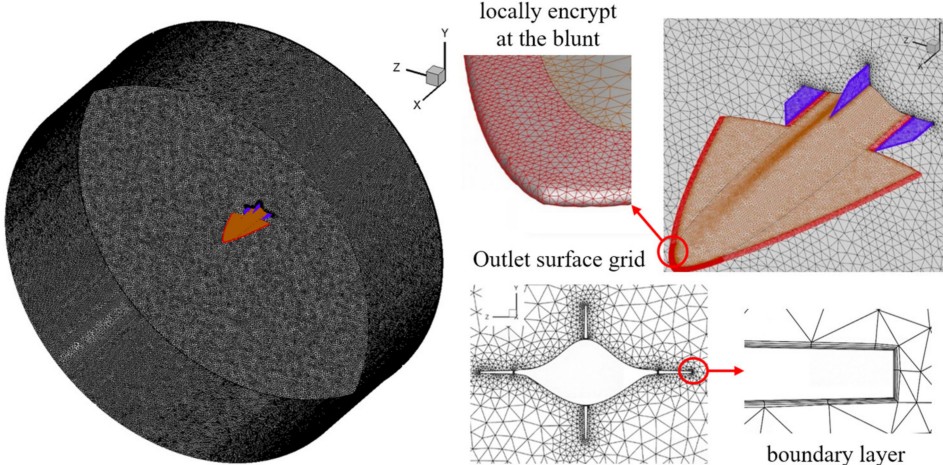

**Figure 8.** Mesh generation.

To ensure the accuracy of the numerical simulation, validation of the turbulence model was conducted using the Ames all-body model from NASA's Ames Research Center [34]. This model has undergone comprehensive wind tunnel testing, and currently serves as a standard for validating turbulence models in the majority of hypersonic vehicle CFD research efforts.

The geometrical data for the Ames all-body model are as follows (refer to Figure 9). The plan view displays a delta wing with a leading edge sweep angle of 75° and a total length of 0.9144 m; it features elliptical cross-sections at various axial positions. The model consists of two main sections: the forebody and the aftbody, with the demarcation line between these sections located at two-thirds of the axial length of the model. The elliptical cross-section of the forebody has a major-to-minor axis ratio of 4. The height of the aftbody minor axis decreases gradually from the interface line towards the trailing edge, eventually reaching a height of zero at the trailing edge.

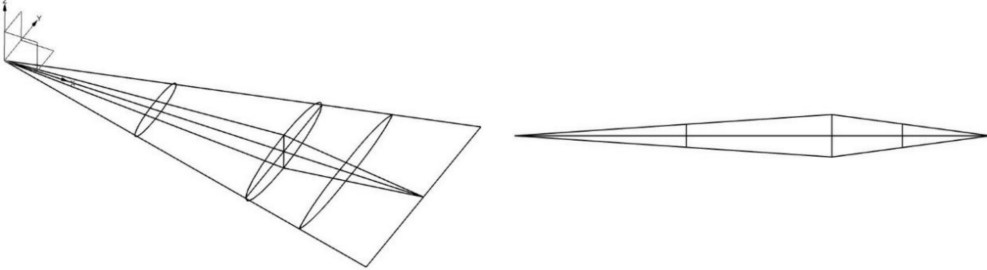

**Figure 9.** Ames all-body model (**left**: 3D view; **right**: left view).

The Ames all-body blowing test conditions are as follows: inflow Mach number: $M_\infty = 7.4$; Reynolds number: $Re_{\infty,L} = 15 \times 10^6$ (L = 0.9144 m); AOA = 0°, 5°, 10°, 15°; inflow temperature: $T_\infty = 62$ K; wall temperature: $T_w = 300$ K.

Figure 10 presents a comparison of the static pressure distribution along the windward/leeward centerline based on numerical simulations using the k-ε turbulence model, calculations from the UPS code developed by the Ames Center, and wind tunnel data provided by the Ames Center. The scattered points represent experimental results, while

the curves depict numerical simulation outcomes. The findings indicate the following: (1) The numerical simulation accurately predicts static pressure in regions where the airfoil profile changes smoothly. However, at locations where the airfoil undergoes a transition (i.e., x = 0.6–0.7 m), the numerical simulation results are slightly lower than the experimental data. (2) The numerical simulation and wind tunnel test data exhibit good agreement at small AOA (<10°). At higher AOA (>10°), the k-ε turbulence model slightly underestimates the static pressure on the windward side; this is a discrepancy similar to that of the results obtained from the UPS numerical simulation code developed by the Ames Center; the discrepancy falls within an acceptable margin of error. These results affirm the applicability of the numerical simulation method based on the k-ε turbulence model for evaluating the aerodynamic characteristics of hypersonic vehicles.

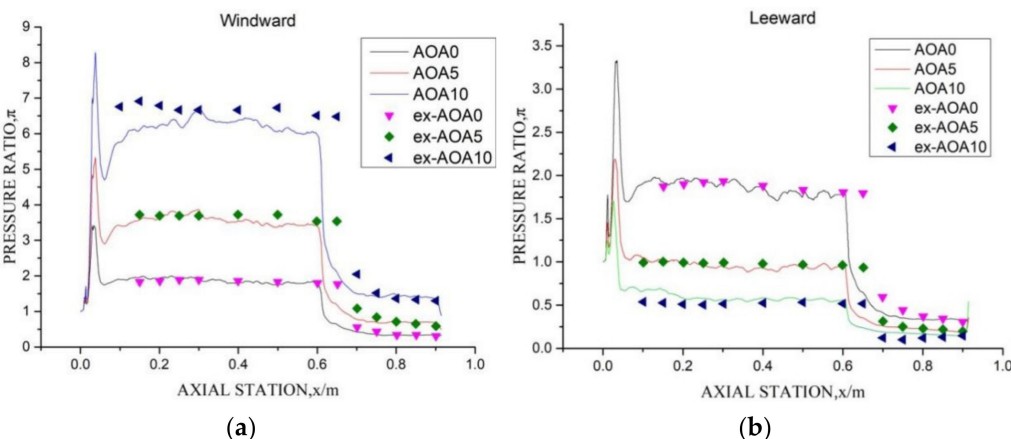

(**a**)  (**b**)

**Figure 10.** The static pressure distribution on the centerline of the Ames all-body model: (**a**) windward side, (**b**) leeward side.

Next, the grid independence verification was conducted using the fully deployed wing model as the baseline. The freestream conditions were set to $M_\infty = 7.0$, $H = 20$ km, and $AOA = 5°$. The thickness of the boundary layer mesh was maintained, and the computed results are presented in Figure 10, where $F_x$ and $F_y$ represent the forces acting on the vehicle in the x and y directions, respectively. All of the forces are determined in the body-fixed coordinate system, and L/D signifies the lift-to-drag ratio. In Table 2, the result of the coarse grid is quite different from those of the medium and fine grids, and the degree of agreement is low, while the results of the medium and fine grids are similar; so, the medium density (7.51 million) grid was used for the subsequent simulation. Employing the same parameters for grid division in the retracted wing model resulted in a grid of 6.14 million cells.

**Table 2.** Grid independence analysis ($M_\infty = 7.0$, $H = 20$ km, AOA = 5°).

| Type | Mesh Number | $F_x$/kN | $F_y$/kN | Lift-to-Drag Ratio |
|---|---|---|---|---|
| Coarse grid | 5.3 million | −11.82 | 101.1 | 4.842 |
| Medium grid | 7.5 million | −11.51 | 99.7 | 4.878 |
| Fine grid | 9.6 million | −11.42 | 98.6 | 4.869 |

Calculations for five distinct inflow conditions were performed using ANSYS CFX, across a range of attack angles from 0 to 10 degrees; these calculations are detailed in Table 3. The chosen inflow Mach numbers varied from Mach 2.5 to 8.5, with the flight altitudes spanning from 6 to 25 km, under the assumption of an ideal gas flow, covering a comprehensive set of 55 state points. Utilizing the geometric parameters of the model depicted in Figure 3, the projected area of the fuselage on the xoz plane was calculated to be 6.077 m$^2$, and it served as the reference area for the vehicle under investigation. All of the wall surfaces were configured to be adiabatic.

**Table 3.** The incoming flow conditions in the vehicle calculation.

| Altitude/km | Temperature/K | Pressure/Pa | Mach |
|---|---|---|---|
| 6 | 249.19 | 47,217.5 | 2.5 |
| 10 | 223.25 | 26,499.76 | 4 |
| 15 | 216.65 | 12,111.4 | 5.5 |
| 20 | 216.65 | 5529.1 | 7 |
| 25 | 221.552 | 2549.15 | 8.5 |

All of the simulation results of the vehicle in the cases of wing deployment and retraction are shown in Appendix A.

### 3.2. Simulation Result and Discussion

#### 3.2.1. Lift/Drag Characteristics

The lift and drag characteristics of the new aerospace vehicle model studied in this study are shown in Figure 11, where the formulas of the lift coefficient and drag coefficient are as follows:

$$C_L = \frac{L}{q \cdot S_{\text{ref}}} \tag{6}$$

$$C_D = \frac{D}{q \cdot S_{\text{ref}}} \tag{7}$$

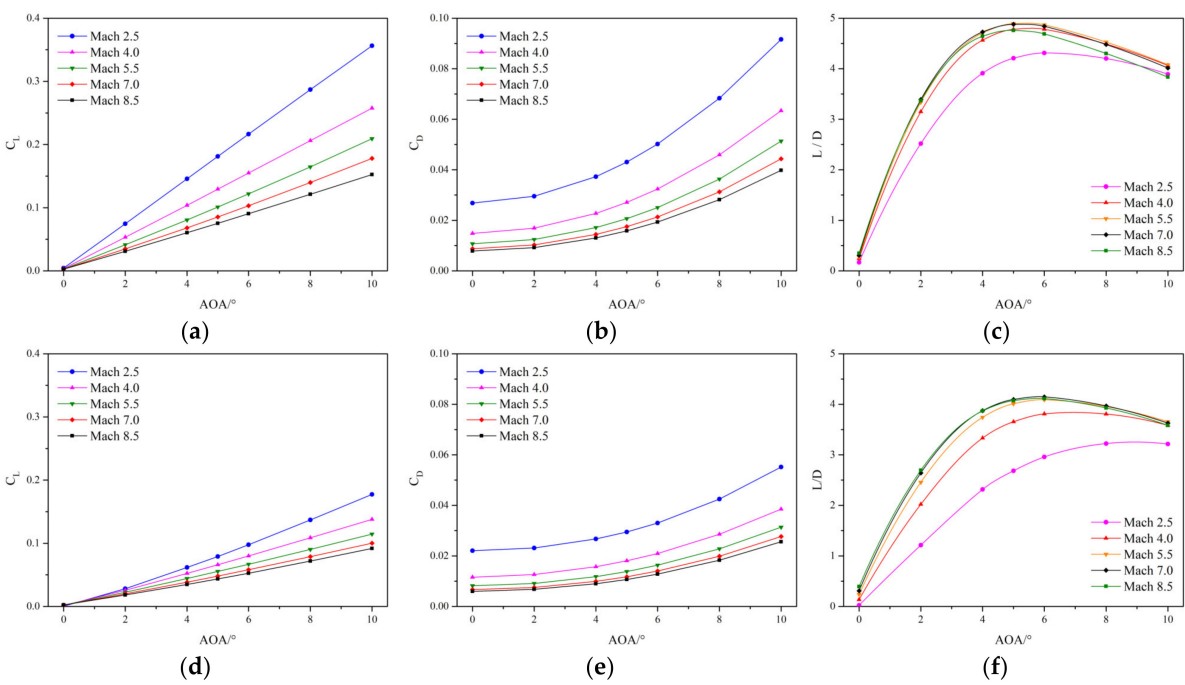

**Figure 11.** Lift resistance characteristics: (**a**) lift coefficient (wings deployed), (**b**) drag coefficient (wings deployed), (**c**) L/D (wings deployed), (**d**) lift coefficient (wings retracted), (**e**) drag coefficient (wings retracted), (**f**) L/D (wings retracted).

In this context, $C_L$ represents lift coefficient and $C_D$ represents the drag coefficient, $L$ means the lift force and $D$ for the drag force, $q$ refers to the dynamic pressure of the free inflow, and $S_{\text{ref}}$ is the reference area.

Comparing the variations in the lift-to-drag characteristics under the same inflow conditions reveals:

1.  The lift coefficient, as illustrated in Figure 11a,d, displays a linear increasing trend with the AOA under various flight conditions and with the wings deployed or retracted. Additionally, it was observed that the lift coefficient increases as the inflow Mach number and flight altitude decrease. At AOA 0°, the lift coefficients of both the deployed and

the retracted configurations are essentially the same; at AOA 5°, for Mach 2.5, the values are 0.181 and 0.079, respectively, indicating that deploying the wings increases the lift by 129% compared to when the wings are retracted; at 10°, the values are 0.356 and 0.177, respectively, representing a 101% increase in lift. This suggests that at Mach 2.5, as the angle of attack increases, the proportion of lift contributed by the wings exhibits a decreasing trend. At Mach 4.0, the lift increases, provided by the wings at AOA 5° and 10° are 96% and 87%, respectively, while at Mach 5.5 and Mach 7.0, these increases stabilize at 82% and 77%. However, at Mach 8.5, the increases drop to 73% and 66%, indicating a declining trend. This illustrates that, as the inflow Mach number and flight altitude increase, the effectiveness of the wings in providing lift gradually decreases.

2. As depicted in Figure 11b,e, the drag coefficient increases in a quadratic trend with the AOA and rises as the inflow Mach number and flight altitude decrease. At AOA 0°, the additional drag when the wings are deployed compared to when the wings are retracted is between 20 and 30%, with the drag coefficient slowly increasing as the inflow Mach number and altitude increase, although its impact is significantly less than that of the lift coefficient. As AOA increases, the drag experienced by the wings also increases; the drag coefficient with deployed wings increases by about 46% at AOA 5° and between 55 and 65% at AOA 10°, but as the inflow Mach number and altitude increase, the drag coefficients for both the deployed and retracted wings show a slow declining trend.

3. The L/D, as depicted in Figure 11c,f, reaches its maximum at AOA 5–6° under different inflow conditions when the wings are deployed, with the maximum L/D between Mach 4.0 and Mach 8.5 consistently at 4.7 and peaking at 4.9 at Mach 5.5. With the wings retracted, the maximum L/D in the high hypersonic range is consistently above 4, showing suboptimal performance in the supersonic range. However, with the assistance of the wings, the L/D at Mach 4.0 can still reach 4.7, although there is a significant decrease in performance at Mach 2.5.

### 3.2.2. Flow Field Analysis (Mach)

To further explore the factors affecting the performance of the vehicle, it is necessary to observe its flow field structure for further analysis. Figure 12 intercepts six cross-sections of the vehicle, starting from the head of the vehicle, to display the Mach number flow field conditions under both deployed and retracted wing modes at Mach 2.5. Figure 12a,b illustrates the Mach number flow fields at AOA 0° with wings retracted and deployed, respectively. Figure 12c,d depicts iso-Mach lines from a side and top view for AOA 0–10° at Mach 2.4, as represented by the red dashed lines, where the airflow velocity inside the iso-Mach lines is lower than that outside the lines.

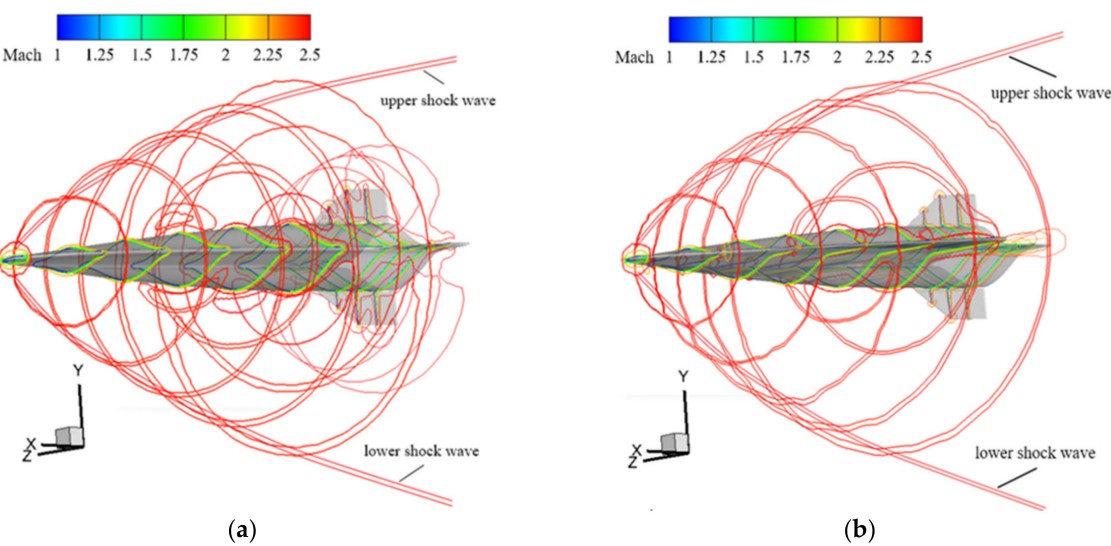

(**a**)            (**b**)

**Figure 12.** *Cont.*

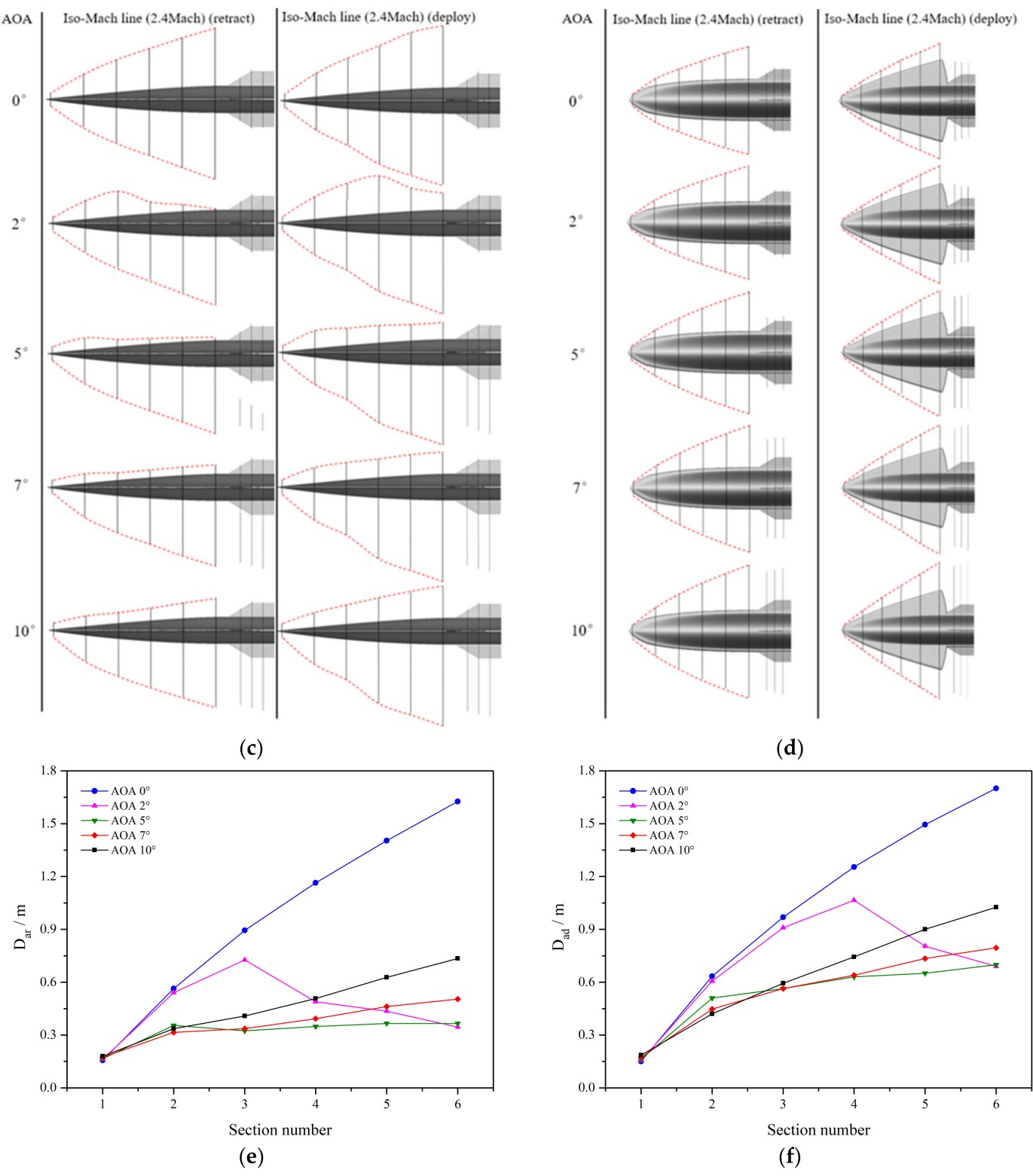

**Figure 12.** *Cont.*

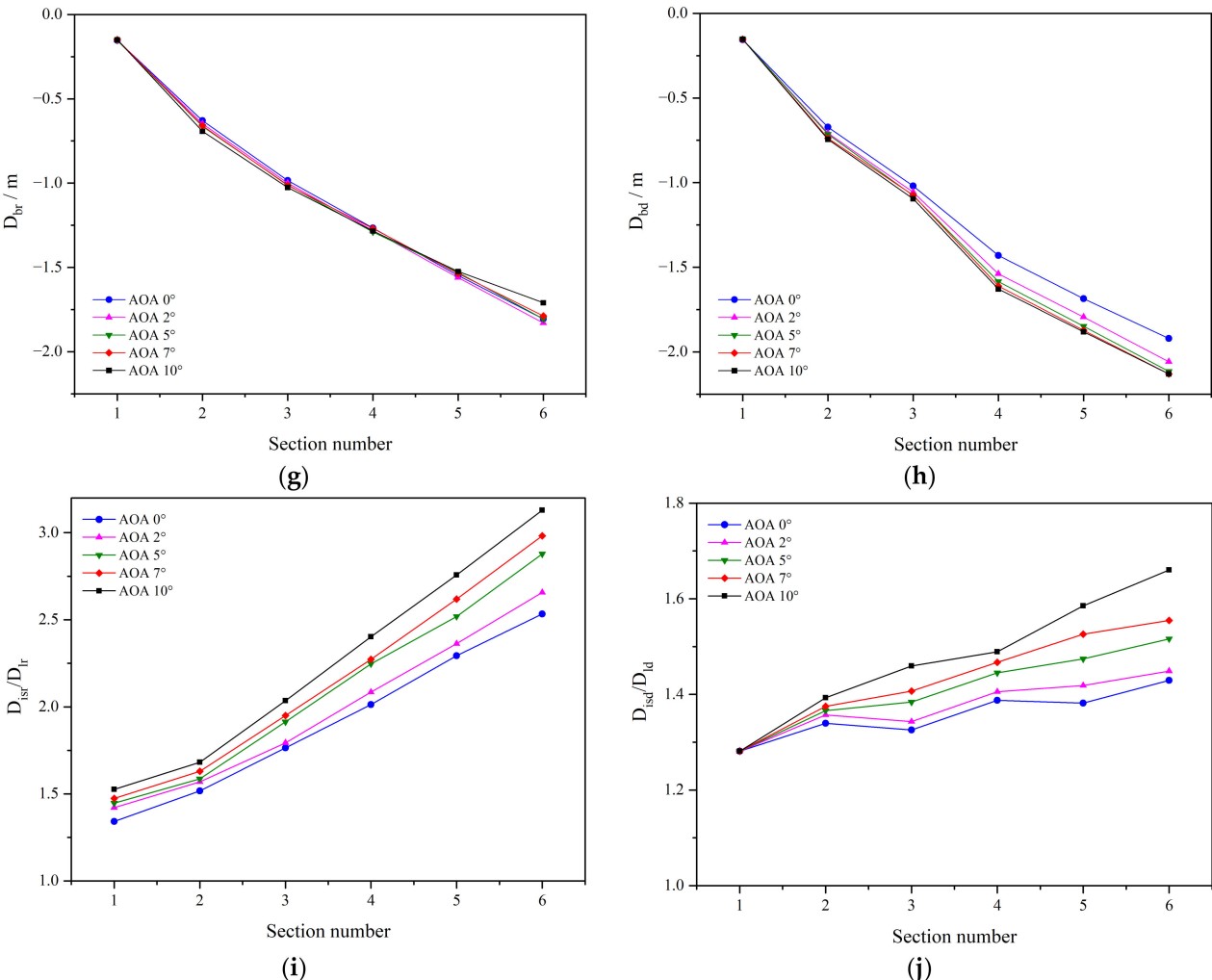

**Figure 12.** The iso-Mach lines diagram of the vehicle under Mach 2.5 conditions: (**a**) wing retracted at AOA 0°, (**b**) wing deployed at AOA 0°, (**c**) side view at AOA 0–10°, (**d**) top view at AOA 0–10°, (**e**) variation of Dar with section number, (**f**) variation of Dad with section number, (**g**) variation of Dbr with section number, (**h**) variation of Dbd with section number, (**i**) variation of Disr/Dlr ratio with section number, (**j**) variation of Disd/Dld ratio with section number.

To highlight the impact of the wing mode and attack angle changes, eight distance parameters were established after measuring the distance from the intersection points of the iso-Mach lines with the six cross-sections to the centerline of the vehicle, as shown in Figure 12c,d; all of the measurements were in meters: $D_{ar}$, $D_{ad}$, $D_{br}$, $D_{bd}$, $D_{isr}$, $D_{lr}$, $D_{isd}$, $D_{ld}$. The parameters $D_{ar}$, $D_{ad}$, $D_{br}$, and $D_{bd}$, which are presented in Figure 12e, f, g, and h, respectively, represent the intersection positions measured up and down the vehicle with wings retracted and deployed, as shown in Figure 12c. These four images clearly display how the iso-Mach lines above and below the fuselage change with the wing modes and angles of attack. $D_{isr}$ and $D_{lr}$ correspond to the distances from the iso-Mach lines and the leading edge line to the centerline of the vehicle with wings retracted, as seen in the top view (Figure 12d). Their $D_{isr}/D_{lr}$ ratio is displayed in Figure 12i; this ratio reflects the distance from the iso-Mach lines to the leading edge under different angles of attack with the wings retracted. $D_{isd}$ and $D_{ld}$ apply to the same situation with the wings deployed (Figure 12j). In addition, Figures 13 and 14 also use these parameters, and their specific values are listed in Appendix B.

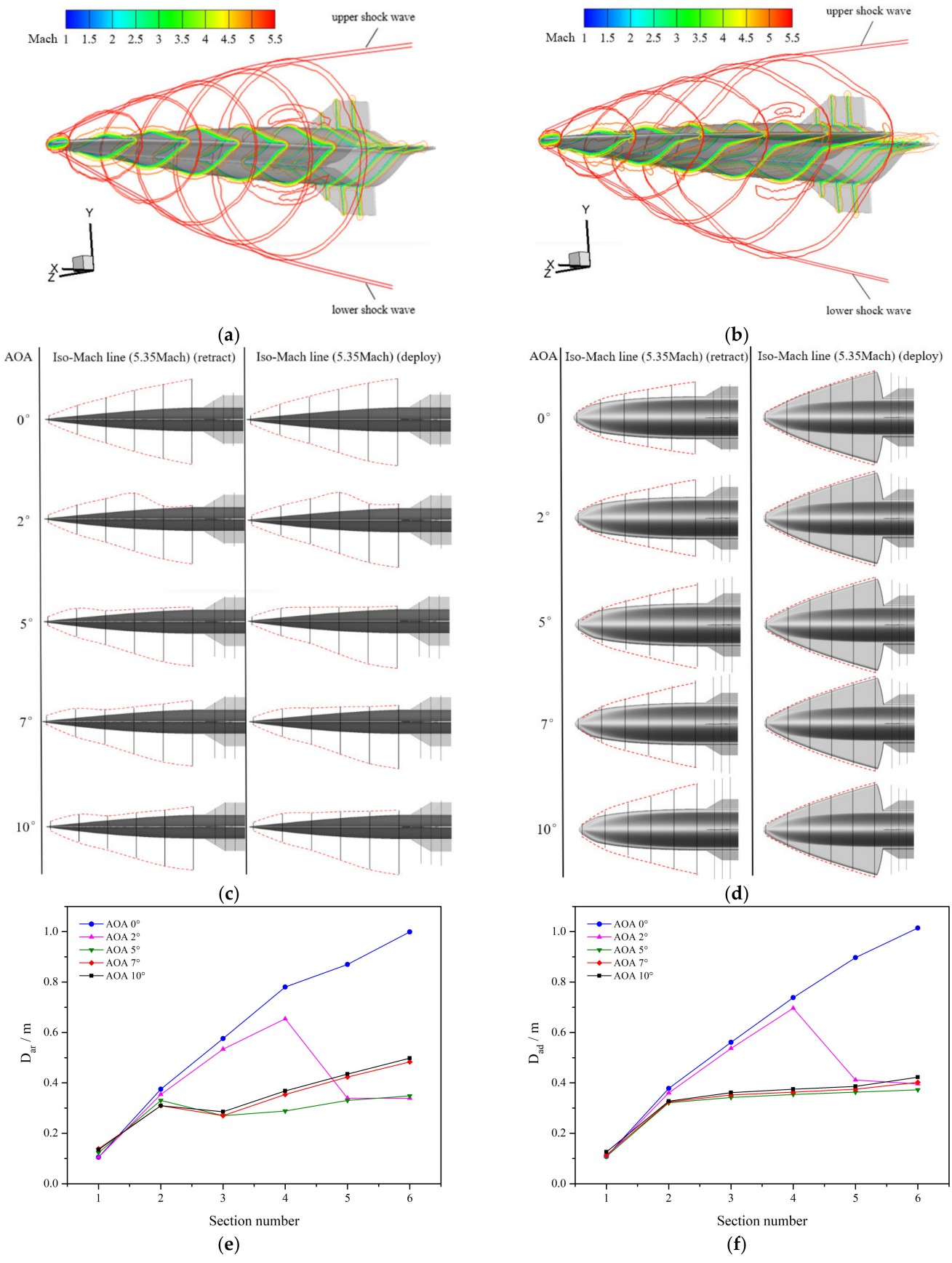

**Figure 13.** *Cont.*

**Figure 13.** The iso-Mach lines diagram of the vehicle under Mach 5.5 conditions: (**a**) wing retracted at AOA 0°, (**b**) wing deployed at AOA 0°, (**c**) side view at AOA 0–10°, (**d**) top view at AOA 0–10°, (**e**) variation of $D_{ar}$ with section number, (**f**) variation of $D_{ad}$ with section number, (**g**) variation of $D_{br}$ with section number, (**h**) variation of $D_{bd}$ with section number, (**i**) variation of $D_{isr}/D_{lr}$ ratio with section number, (**j**) variation of $D_{isd}/D_{ld}$ ratio with section number.

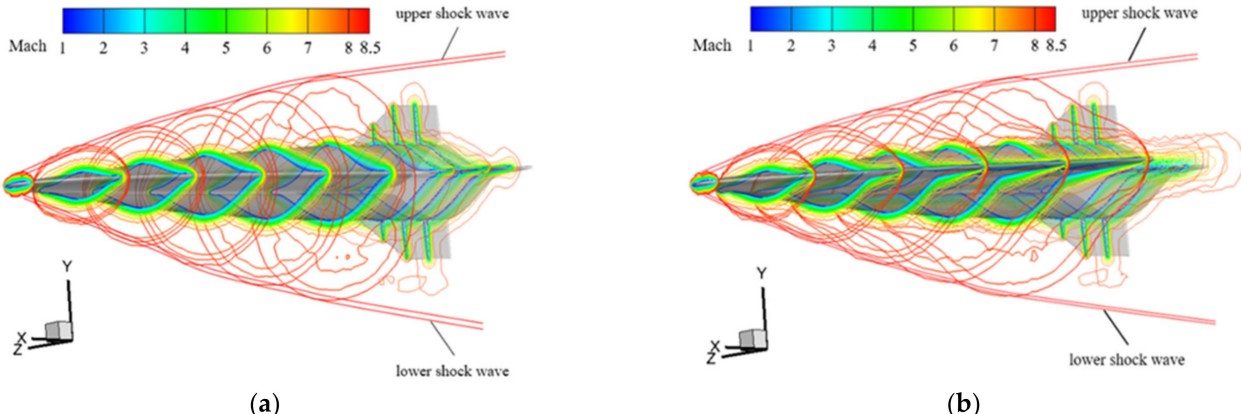

**Figure 14.** *Cont.*

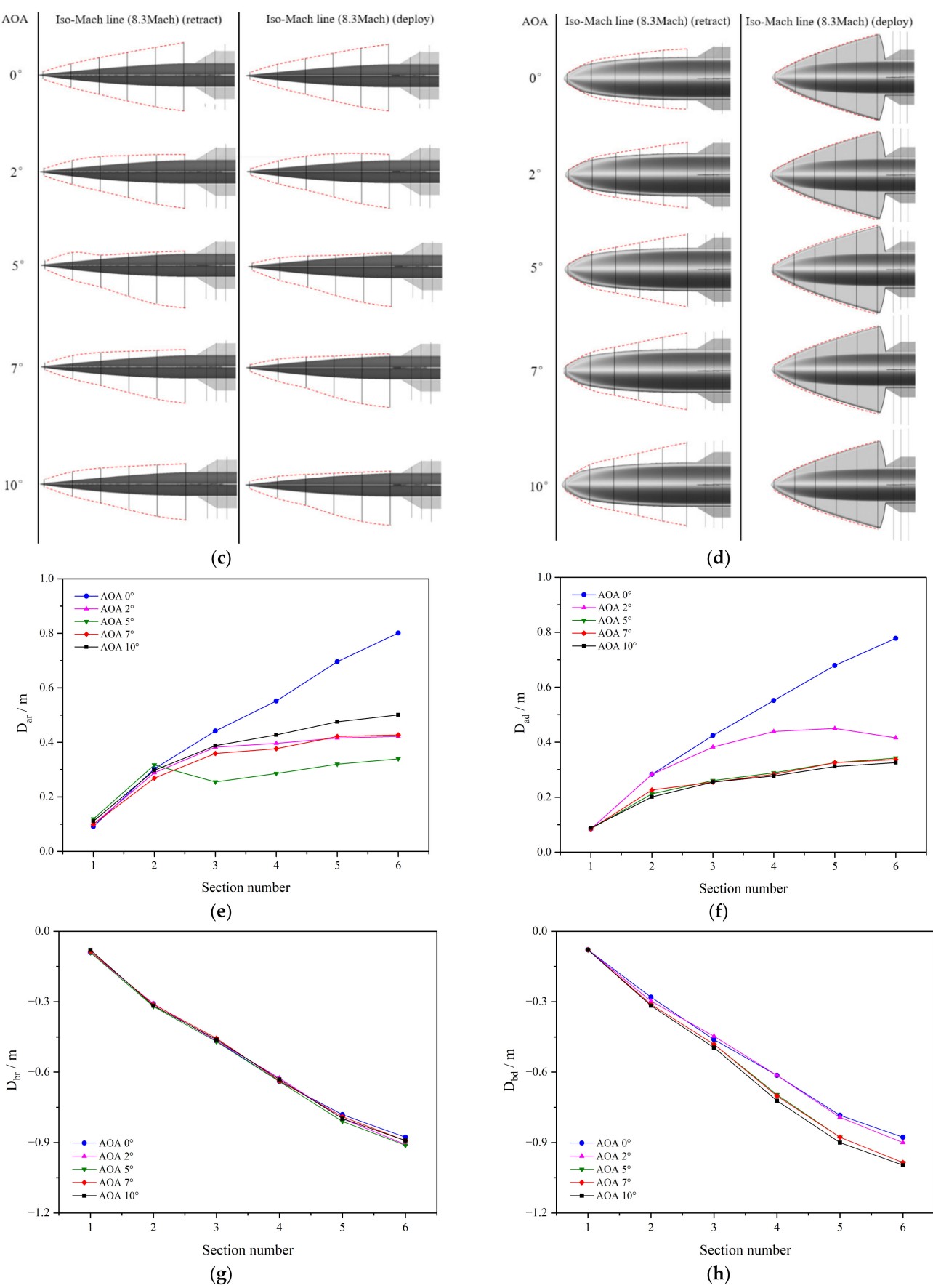

**Figure 14.** *Cont.*

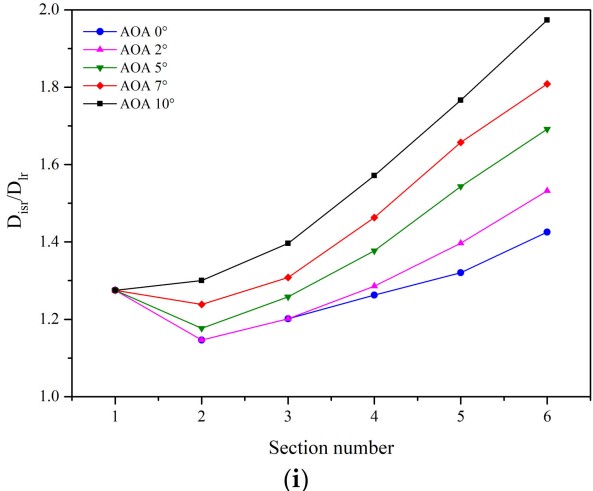
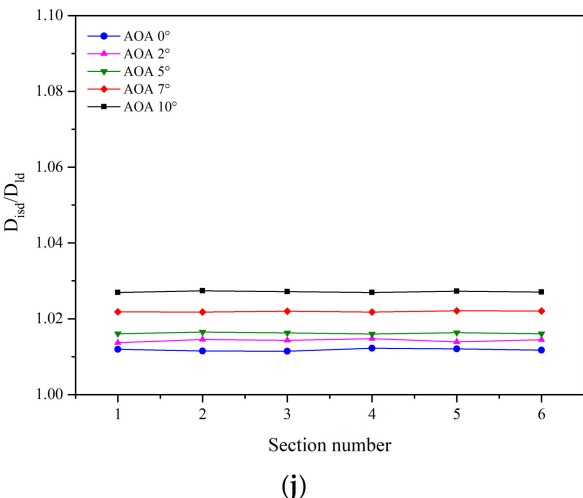

(**i**)                                               (**j**)

**Figure 14.** The iso-Mach lines diagram of the vehicle under Mach 8.5 conditions: (**a**) wing retracted at AOA 0°, (**b**) wing deployed at AOA 0°, (**c**) side view at AOA 0–10°, (**d**) top view at AOA 0–10°, (**e**) variation of $D_{ar}$ with section number, (**f**) variation of $D_{ad}$ with section number, (**g**) variation of $D_{br}$ with section number, (**h**) variation of $D_{bd}$ with section number, (**i**) variation of $D_{isr}/D_{lr}$ ratio with section number, (**j**) variation of $D_{isd}/D_{ld}$ ratio with section number.

From Figure 12, the following can be concluded:

1.  Figure 12c describes the changes in iso-Mach lines above and below the fuselage with wing mode, and Figure 12e–h provides detailed information on the positions of the intersections between the iso-Mach lines and the six cross-sections. These images collectively reveal that the iso-Mach lines are further from the fuselage at various attack angles when the wings are deployed.

2.  The iso-Mach line above the fuselage at AOA 5° is the lowest for both wing modes, and as seen in Figure 11, this condition yields the maximum L/D for the vehicle; at AOA 2° and 5°, a turning appears on the upper surface iso-Mach lines due to the upper surface gradually transitioning from the windward to the leeward side, with gas decelerating through the shock wave, then accelerating over the surface through the expansion wave; with the increasing attack angle, the expansion waves gradually move forward. Subsequently, at AOA 7° and 10°, the iso-Mach lines gradually rise due to an increased overflow with the rising attack angle, causing increased air pressure above the fuselage and, by the principle of the conservation of mechanical energy, an increase in pressure of the potential energy means a certain decrease in the velocity of the potential energy; hence, the iso-Mach line moves up, enlarging the low-speed area.

3.  In Figure 12g,h, a downward turn is observed in the iso-Mach lines at the third and fourth cross-sections with the wings deployed, compared to when the wings are retracted. This indicates that deploying the wings increases the pressure on the lower side of the vehicle, which is the same as the aforementioned principle of the conservation of mechanical energy, resulting in an expansion of the low-speed area below.

4.  Upon examining Figure 12d,i,j, it becomes evident that at Mach 2.5, the angle at which the wings are deployed is less than that of the shock wave angle, resulting in an overflow at the leading edges of the wings. Consequently, at Mach 2.5, the vehicle lacks the capability for the waverider effect, leading to a maximum L/D that is inferior to that observed at other Mach numbers. Moreover, Figure 12i,j demonstrates that, with an increase in the attack angle, the iso-Mach lines progressively distance themselves from the leading edge of the vehicle, illustrating an augmentation in the overflow on both sides of the vehicle as the attack angle increases.

Figure 13 shows the conditions at Mach 5.5, with Mach 5.35 selected as the iso-Mach lines. From the Figure 13, we can deduce:

1.  As shown in Figure 13c,e,f,g,h, the iso-Mach lines above the fuselage at AOA 5°, which are consistent with those of Mach 2.5, are the lowest with both wing modes; a turn also appears in the upper surface iso-Mach lines at AOA 2° and 5°. Subsequently, at AOA 7° and 10°, the iso-Mach lines with the wings retracted begin to rise again, but the rise is considerably less significant with the wings deployed due to the shock waves being closer to the wing-leading edges, suppressing the overflow from the lower to the upper surface, which confirms the occurrence of the waverider effect.

2.  In Figure 13g,h, compared to when the wings are retracted, the iso-Mach lines under the lower side of the vehicle with the wings deployed show a slight downward shift overall. This shift can be attributed to the generation of the waverider effect at Mach 5.5, which results in a higher airflow pressure below and leads to an expansion of the low-speed area; this is consistent with the aforementioned principle of the conservation of mechanical energy.

3.  It can be seen in Figure 13d,i,j, at Mach 5.5 with the wings deployed, that the iso-Mach lines are very close to the wing-leading edges, confirming the occurrence of the waverider effect, with the maximum L/D under this inflow condition reaching the highest point compared to the other Mach inflow conditions.

4.  As depicted in Figure 13j, with an increase in the attack angle, the distance between the iso-Mach lines and the leading edges of the vehicle incrementally expand, which is consistent with the phenomenon observed at Mach 2.5. It also demonstrates that, under the Mach 5.5 inflow condition, the overflow on both sides of the vehicle amplifies as the attack angle ascends.

Figure 14 also addresses the conditions at Mach 8.5, with Mach 8.3 selected as the iso-Mach lines:

1.  As shown in Figure 14c,e,f,g,h, the iso-Mach line above the fuselage at AOA 5° is the lowest in both wing modes, which is consistent with Mach 2.5 and 5.5; a turn appears in the upper surface iso-Mach lines at AOA 2° and 5°. Subsequently, at AOA 7° and 10°, the iso-Mach lines above the fuselage with the wings retracted have a much larger rise than the fuselage with the wings deployed; this is consistent with Mach 5.5 and confirms the occurrence of the waverider effect at Mach 8.5.

2.  In Figure 14g,h, comparing wings deployed with wings retracted, there is no significant turn in the iso-Mach lines on the lower side of the vehicle, unlike at Mach 2.5 and 5.5. Although waverider effects are still generated, the faster inflow speed has a greater impact on the shock waves, making this phenomenon less pronounced.

3.  As can be seen in Figure 14d,i,j, at Mach 8.5 with wings deployed, the iso-Mach lines are closer to the wing-leading edges compared to Mach 5.5, confirming the occurrence of the waverider effect, but due to an increased wave drag, the highest L/D at Mach 8.5 is slightly lower than at Mach 5.5.

4.  From Figure 14j, as the attack angle increases, the distance from the iso-Mach lines to the leading edge of the vehicle slightly increases, but to a lesser extent than at Mach 5.5; combined with the conditions at Mach 2.5, this indicates that the closer the shock waves are to the leading edge of the vehicle, the smaller the overflow from the lower to the upper surface.

Comparing the three sets of Figures 12–14, we summarize as follows:

The subfigures e and f in Figures 12–14 show the distance from the iso-Mach lines above the vehicle to the centerline of the vehicle with the wings retracted and deployed. For the $D_{ar}$ and $D_{ad}$ values at Mach 2.5 (Figure 12e,f) and the $D_{ar}$ values at Mach 5.5 and Mach 8.5 (subfigures e in Figures 13 and 14), the position of the iso-Mach lines at AOA 5° are closest to the fuselage. From AOA 0 to 5°, as the attack angle increases, the upper surface gradually transitions from the windward to the leeward side and experiences expansion waves. The airflow over the upper surface first decelerates through the shock

waves and then accelerates through the expansion waves; hence, the iso-Mach lines in the front half of the fuselage show a 'low-high-low' shape. After AOA 5°, the overflow of air from the lower to the upper surface increases, causing the iso-Mach lines to move up. When the wings are deployed, displaying the waverider effect seen for $D_{ad}$ at Mach 5.5 and 8.5 (subfigures f in Figures 13 and 14), the iso-Mach lines do not significantly rise between AOA 5° and 10°.

Subfigures i and j in Figures 12–14 show the relationship between the iso-Mach lines and the leading edge of the vehicle in the top view, with higher Mach numbers bringing the iso-Mach lines closer to the leading edge of the vehicle. This also means that, for future designs of the hypersonic variable-configuration vehicle, adjusting the sweep of the wings to match the shock wave angle at different speeds could generate a waverider effect, enhancing the vehicle's aerodynamic performance.

### 3.2.3. Flow Field Analysis (Pressure)

The aforementioned analysis of the effect of Mach number variations on the flow field of the vehicle under different inflow conditions sets the stage for a deeper examination of the flow field conditions when the L/D performance is optimal. Figure 15, with the nose tip of the vehicle as the origin, establishes four cross-sections perpendicular to the inflow direction. The distances of these cross-sections from the origin are 0.3 m, 1.4 m, 2.5 m, and 3.6 m, respectively. The pressure ratio, $P/P_\infty$, is used to display the pressure conditions at AOA 5° when the wings are deployed (left side) versus when they are retracted (right side) under different inflow conditions. (In Figure 15, for the convenience of comparison, the two subplots (the wings deployed and retracted modes) are put together and are not the result of a configuration with one wing deployed and the other retracted. The same logic applies to Figures 16 and 17).

From Figure 15, the following conclusions can be drawn:

1. As the inflow Mach number increases, the pressure ratio on the lower surface progressively strengthens, which also represents an increase in the intensity of the shock wave.
2. At Mach 2.5, the inflow pressure at each cross-section is relatively low. Regardless of whether the wings are deployed or retracted, overflow from the lower to the upper surface occurs, indicating that the vehicle does not exhibit a high lift-to-drag performance under this inflow condition; as shown in Figure 11c, the maximum lift-to-drag ratio is 4.3, which is lower than that of the other inflow conditions.
3. At Mach 4.0, the front of the vehicle (0.3 m) shows a certain waverider effect, enhancing the lift-to-drag performance. However, in subsequent parts of the vehicle, significant overflow is observed when the wings are retracted, while deploying the wings substantially suppresses overflow on the lower surface and significantly improves the lift-to-drag performance.
4. At Mach 5.5, due to the reduction in the shock wave angle, a high-pressure zone begins to appear at the leading edge of the vehicle. As shown by the closed lift-to-drag ratio in Figure 11f, the waverider effect is noticeably enhanced. With the wings deployed, a high-pressure zone appears at the edges in all four sections, trapping high-pressure air beneath the lower surface and achieving the highest L/D of 4.9.
5. At Mach 7.0 and Mach 8.5, with the wings retracted, the first three sections (0.3 m, 1.4 m, and 2.5 m) display the waverider effect, showing a higher L/D compared to the first three conditions (Mach 2.5, 4.0, and 5.5). When the wings are deployed, the high-pressure area at the wing edges gradually expands, resulting in a higher wave drag and thus slightly reducing the lift-to-drag performance when compared to Mach 5.5.

Corresponding to Figure 15, Figure 16 shows the distribution of the pressure ratio and wall streamlines on the upper and lower surfaces at AOA 5° under various inflow Mach numbers.

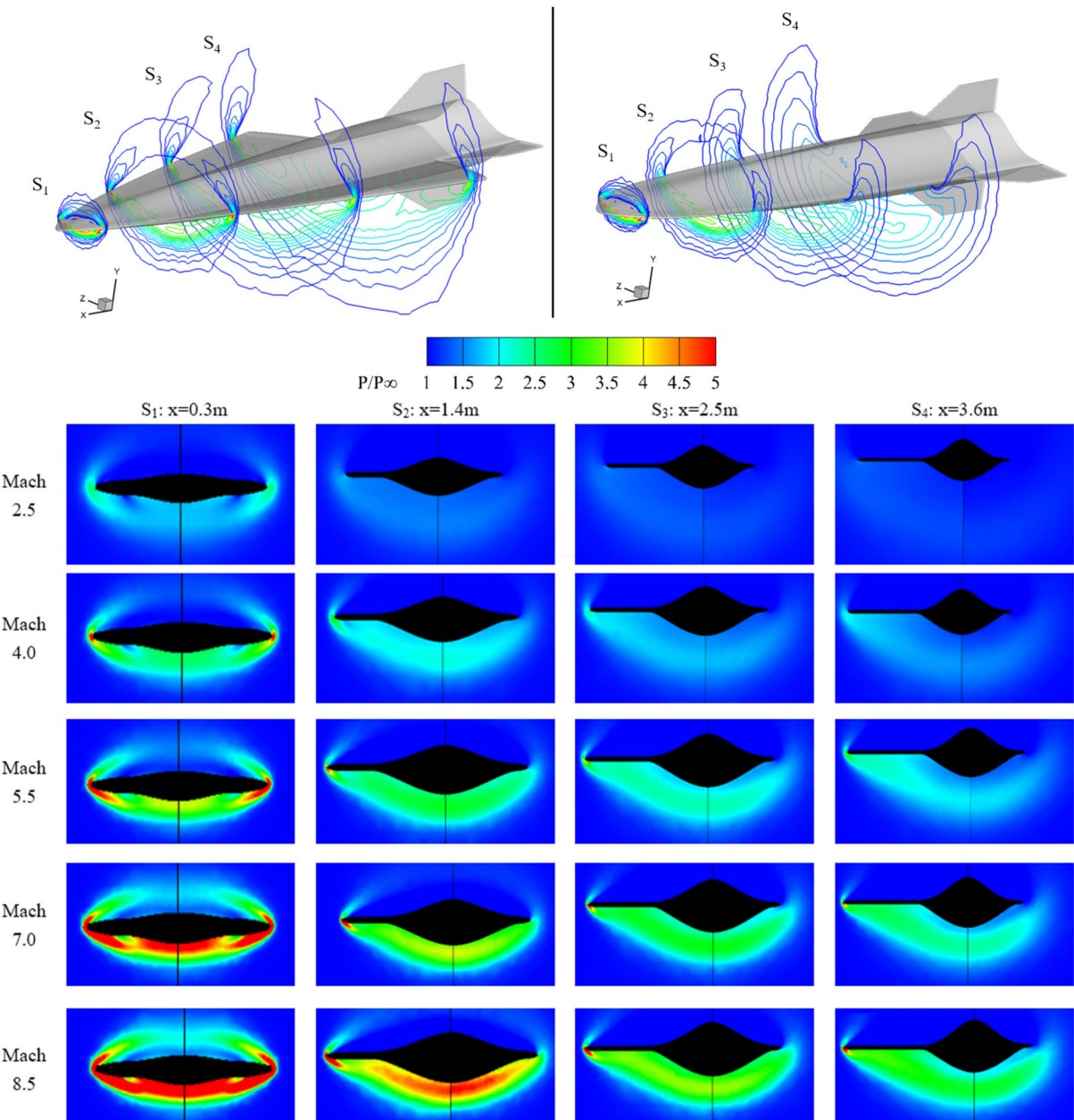

**Figure 15.** The pressure ratio at Mach 2.5–8.5 and AOA 5° on four cross-sections of the vehicle.

The Figure 16 reveals that, as the inflow Mach number increases, the pressure ratio at the front of both the lower and upper surfaces also increases. The wall streamlines on the lower surface slightly diverge outward, while those on the upper surface of the fuselage move closer to the centerline. Initially, the wall streamlines at the leading edge of the wings align with the direction of overflow, gradually becoming parallel to the inflow direction.

At Mach 2.5, the pressure ratio distribution on the lower surface is relatively consistent whether the wings are deployed or retracted, indicating the absence of the waverider effect. In contrast, for the following four Mach numbers, the pressure ratio distribution on the lower surface when the wings are deployed is higher than when retracted, signifying the occurrence of the waverider effect. Correspondingly, the highest L/Ds under these four Mach conditions are all above 4.7.

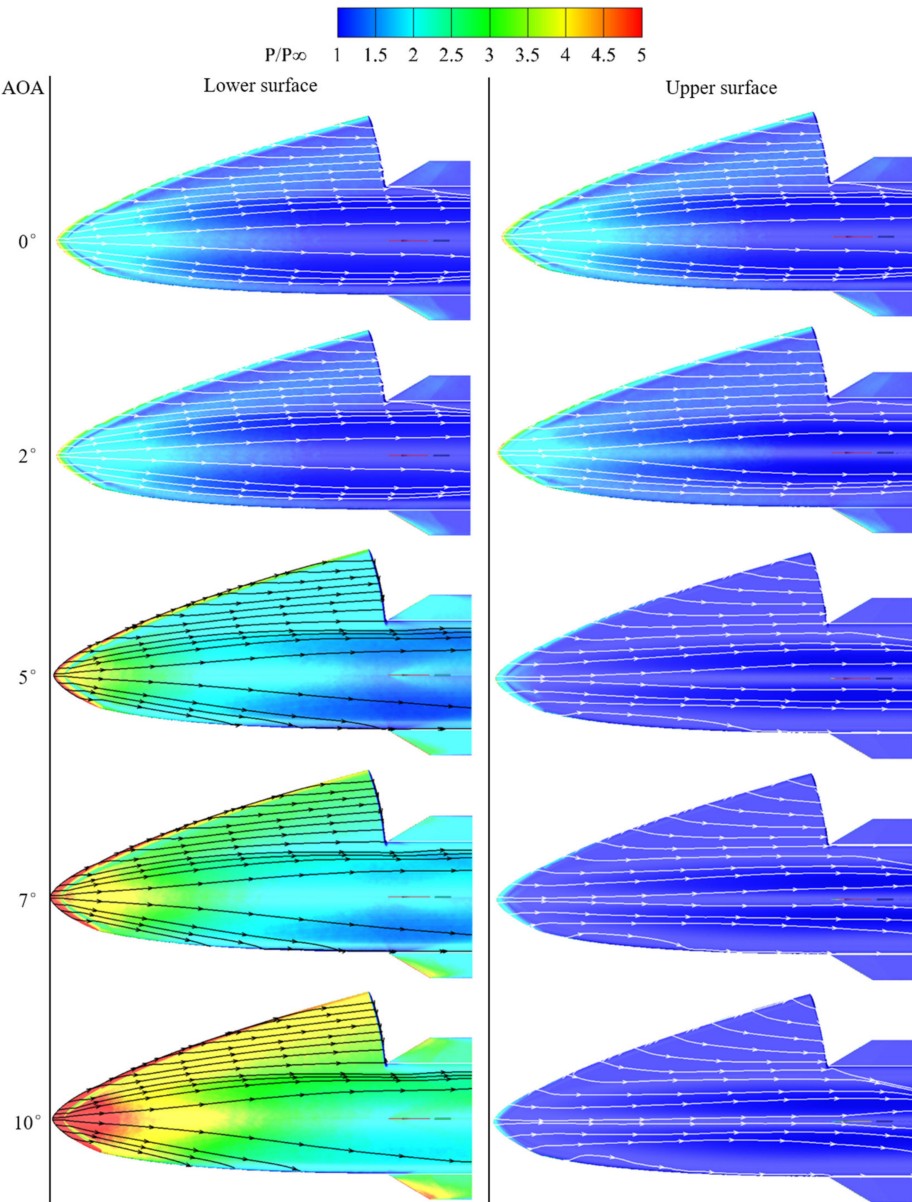

**Figure 16.** The pressure ratio at AOA 5° on the upper and lower surfaces of the vehicle.

For the flight conditions at Mach 5.5, where the lift-to-drag performance is optimized, the pressure ratio contours of the upper and lower surfaces of the vehicle are presented for AOA 0–10° (see Figure 17). From the Figure 17, the following observations can be made:

1.  As the AOA increases, the pressure on the lower surface gradually rises, and the flow lines disperse towards the sides; conversely, the pressure on the upper surface decreases, with the wall flow lines converging towards the centerline.
2.  At AOA 0°, the pressure is concentrated around the nose, the leading edge, and the central line on the fuselage. Due to the symmetrical blunt leading edge, the pressure distribution at the leading edge is almost identical on both the upper and lower surfaces. As shown in the sections (see the S1 sections in Figure 16), the lower surface of the fuselage exhibits more protrusion than the upper surface; hence, there is a higher pressure distribution.
3.  For the upper surface at a certain AOA during the flight, the airflow forms a small high-pressure area after passing the leading edge shockwave, then continuously expands and accelerates along the upper surface, resulting in an overall low-pressure area.

4. Corresponding to the section at Ma 5.5 depicted in Figure 16, with the wings retracted, the angle between the leading edge tangent line of the fuselage and the direction of the incoming flow exceeds the shock wave angle. It happens in the area of approximately the first third of the lower surface. This causes the high-pressure airflow below to concentrate on the lower surface due to the combined effects of the shockwave and leading edge, thereby generating a certain waverider effect in this area, which led to high pressure, as the Figure 16 shows. However, beyond this area, the angle gradually reduces to less than the shock wave angle, and the gas under the lower surface overflows upwards at the middle of the fuselage, thus reducing pressure in this area and leading to some lift loss. At this juncture, deploying the wings in order to have the leading edge of the wings take over from the leading edge of the fuselage can re-establish the waverider effect. This adjustment not only elevates the pressure in the previously low-pressure area observed during wing retraction around the midsection of the fuselage but also, with the added lift provided by the wing's lower surface, optimizes the L/D of the vehicle under this inflow condition.

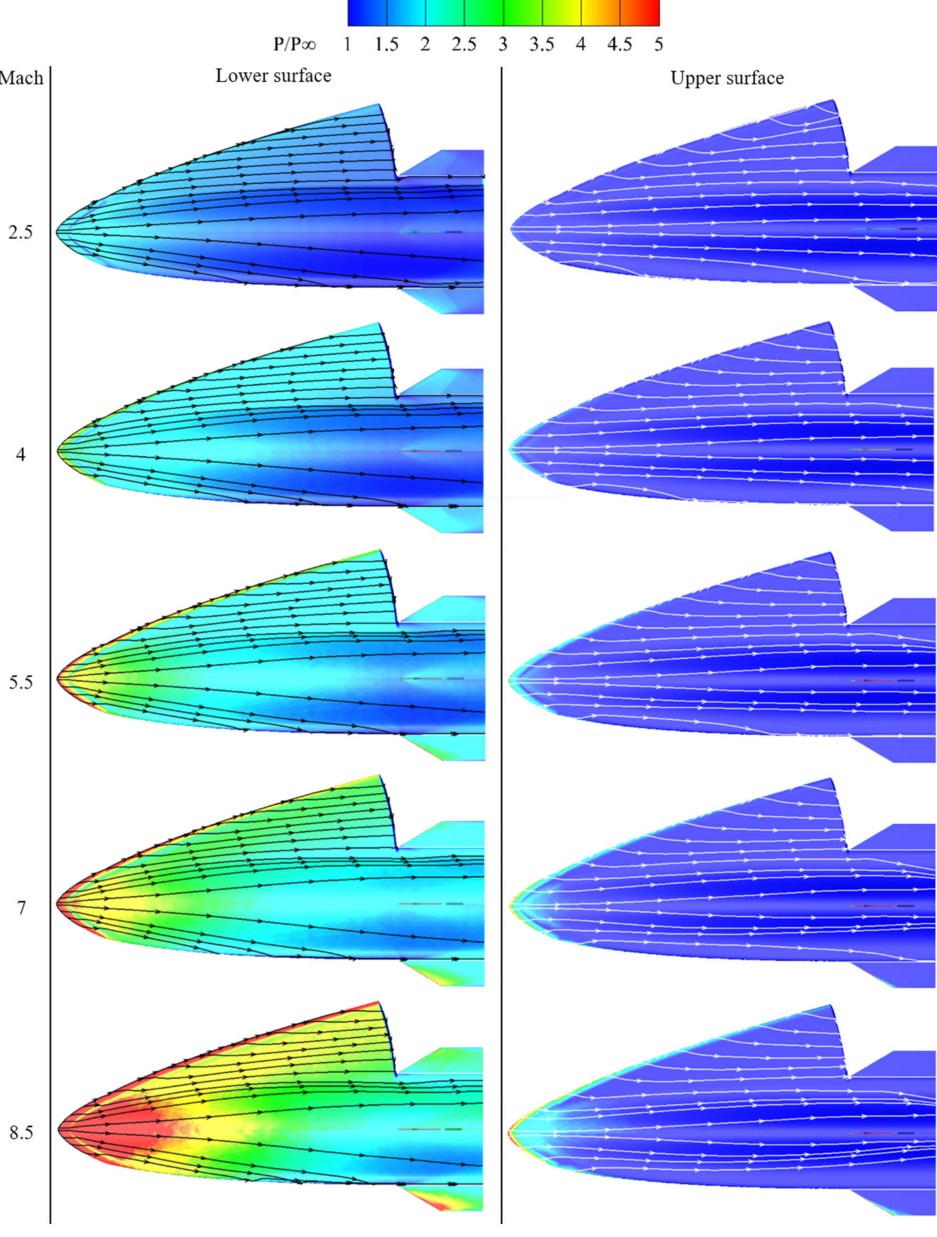

**Figure 17.** The pressure ratio at Mach 5.5 on the upper and lower surfaces of the vehicle.

## 4. Conclusions

This study proposed an aerodynamic configuration for a wide-range, wing-morphing vehicle and achieved a parametric design and shape optimization of the entire vehicle based on UG secondary development technology. The design results obtained through the aforementioned methods are numerically simulated to study their characteristics, yielding the following conclusions:

1. The parametric design method in this study, which was based on two-dimensional B-splines, employed 27 parameters to fully characterize the features of the entire vehicle. This study also presented a shape optimization method suitable for internal payloads of any number and shape. Both of the methods were implemented through UG secondary development, allowing for the rapid design and optimization of hypersonic vehicle aerodynamic shapes according to specified internal payloads in practical engineering applications, thereby enhancing design efficiency.

2. The numerical simulations of the models generated using the above methods across a typical Mach number range (Mach 2.5–8.5) analyzed the lift/drag characteristics of the wings in various flight vectors. The results demonstrated that the highest L/D exceeded 4.7 under flow conditions from Mach 4 to Mach 8.5. With the wings retracted, the drag coefficient remained below 0.02 within AOA 5°, reducing drag by approximately 20–30% compared to the scenario with the wings deployed. Furthermore, a continuous and flexible adjustment of the L/D (0.3–4.7) within a 0–5° angle of attack was achievable, endowing the vehicle with a certain maneuverability and flexibility.

3. Flow field analysis of the vehicle in various flight states showed that between Mach 2.5 and Mach 8.5 the vehicle can match the corresponding shockwave angles at different inflow velocities by adjusting the wing opening, thereby generating a certain waverider effect and minimizing overflow from the lower surface to the upper surface. Analysis of the pressure field indicated that wing deployment maintains a higher pressure difference between the lower and upper surfaces, effectively enhancing overall lift performance.

**Author Contributions:** Conceptualization, W.Z. and Z.Y.; methodology, Z.Y.; software, Z.J.; validation, F.M.; formal analysis, Z.J.; investigation, W.Z., J.C. and Z.Y.; data curation, Z.J.; writing—original draft preparation, Z.J. and F.M.; writing—review and editing, Z.Y., X.H., O.M. and Y.L.; visualization, Z.J.; supervision, Z.Y. and Y.L.; funding acquisition, W.Z., J.C. and Z.Y. All authors have read and agreed to the published version of the manuscript.

**Funding:** This research is supported by CALT Funding (CALT2023-07); NSFC Research Fund for International Young Scientists under Grant No. 52350410466.

**Institutional Review Board Statement:** Not applicable.

**Informed Consent Statement:** Not applicable.

**Data Availability Statement:** Data are contained within the article.

**Conflicts of Interest:** The authors declare no conflicts of interest.

## Appendix A. Simulation Results

The force performance of the vehicle in the numerical simulation are listed in Tables A1 and A2, including the forces acting on the vehicle in the x and y directions ($F_x$ and $F_y$), lift force, drag force, L/D, lift coefficient ($C_L$), and drag coefficient ($C_D$).

**Table A1.** The force characteristics of the vehicle with the wings retracted under different incoming flow conditions.

| Mach | AOA/° | $F_x$/N | $F_y$/N | Lift Force/N | Drag Force/N | L/D | $C_L$ | $C_D$ |
|---|---|---|---|---|---|---|---|---|
| 8.5 | 0 | −4663.55 | 1828.79 | 1828.79 | 4663.55 | 0.392145 | 0.002334 | 0.005953 |
| 8.5 | 2 | −4790.66 | 14,427.50 | 14,251.52 | 5291.25 | 2.693411 | 0.018191 | 0.006754 |
| 8.5 | 4 | −5115.03 | 27,615.60 | 27,191.52 | 7028.94 | 3.868512 | 0.034708 | 0.008972 |
| 8.5 | 5 | −5356.08 | 34,609.50 | 34,010.99 | 8352.12 | 4.072141 | 0.043412 | 0.010661 |
| 8.5 | 6 | −5644.95 | 41,933.90 | 41,114.12 | 9997.31 | 4.112518 | 0.052479 | 0.012761 |
| 8.5 | 7 | −5976.00 | 49,695.90 | 48,597.18 | 11,987.86 | 4.053867 | 0.06203 | 0.015301 |
| 8.5 | 8 | −6355.64 | 57,731.10 | 56,284.73 | 14,328.40 | 3.928193 | 0.071843 | 0.018289 |
| 8.5 | 10 | −7274.43 | 74,260.21 | 71,868.84 | 20,059.07 | 3.582861 | 0.091735 | 0.025604 |
| 7 | 0 | −7678.78 | 2375.10 | 2375.10 | 7678.78 | 0.309307 | 0.002061 | 0.006663 |
| 7 | 2 | −7861.00 | 23,178.80 | 22,890.34 | 8665.14 | 2.641658 | 0.019863 | 0.007519 |
| 7 | 4 | −8316.83 | 44,992.10 | 44,302.35 | 11,435.06 | 3.874256 | 0.038443 | 0.009923 |
| 7 | 5 | −8652.98 | 56,409.50 | 55,440.69 | 13,536.46 | 4.095655 | 0.048108 | 0.011746 |
| 7 | 6 | −9053.81 | 68,234.30 | 66,914.12 | 16,136.64 | 4.14672 | 0.058064 | 0.014002 |
| 7 | 7 | −9513.92 | 80,532.86 | 78,773.12 | 19,257.49 | 4.090518 | 0.068355 | 0.01671 |
| 7 | 8 | −10,041.30 | 93,197.40 | 90,892.93 | 22,914.15 | 3.966673 | 0.078871 | 0.019884 |
| 7 | 10 | −11,313.89 | 119,262.35 | 115,485.85 | 31,851.69 | 3.625737 | 0.100212 | 0.027639 |
| 5.5 | 0 | −12,712.10 | 2991.54 | 2991.54 | 12,712.10 | 0.23533 | 0.00192 | 0.008157 |
| 5.5 | 2 | −12,951.10 | 35,294.00 | 34,820.51 | 14,174.95 | 2.456482 | 0.022344 | 0.009096 |
| 5.5 | 4 | −13,524.86 | 69,819.30 | 68,705.77 | 18,362.26 | 3.741683 | 0.044087 | 0.011783 |
| 5.5 | 5 | −13,920.40 | 87,867.80 | 86,320.19 | 21,525.61 | 4.010116 | 0.05539 | 0.013813 |
| 5.5 | 6 | −14,410.00 | 106,081.47 | 103,994.09 | 25,419.59 | 4.0911 | 0.066731 | 0.016311 |
| 5.5 | 7 | −14,971.63 | 124,917.71 | 122,162.01 | 30,083.68 | 4.060741 | 0.078389 | 0.019304 |
| 5.5 | 8 | −15,620.70 | 144,136.00 | 140,559.30 | 35,528.53 | 3.956237 | 0.090195 | 0.022798 |
| 5.5 | 10 | −17,122.80 | 184,323.00 | 178,549.38 | 48,870.02 | 3.653556 | 0.114572 | 0.031359 |
| 4 | 0 | −20,874.10 | 2798.47 | 2798.47 | 20,874.10 | 0.134064 | 0.001552 | 0.011574 |
| 4 | 2 | −21,126.30 | 46,696.70 | 45,930.96 | 22,743.12 | 2.019554 | 0.025468 | 0.012611 |
| 4 | 4 | −21,669.39 | 96,108.16 | 94,362.47 | 28,320.77 | 3.331917 | 0.052322 | 0.015703 |
| 4 | 5 | −22,054.80 | 121,316.00 | 118,932.15 | 32,544.26 | 3.654474 | 0.065945 | 0.018045 |
| 4 | 6 | −22,531.16 | 147,140.21 | 143,979.01 | 37,788.07 | 3.810171 | 0.079833 | 0.020953 |
| 4 | 7 | −23,078.51 | 173,671.93 | 169,564.84 | 44,071.77 | 3.84747 | 0.09402 | 0.024437 |
| 4 | 8 | −23,705.90 | 201,246.00 | 195,988.26 | 51,483.23 | 3.806837 | 0.108671 | 0.028546 |
| 4 | 10 | −25,184.00 | 256,857.00 | 248,581.61 | 69,404.15 | 3.581653 | 0.137833 | 0.038483 |
| 2.5 | 0 | −27,668.00 | 616.36 | 616.36 | 27,668.00 | 0.022277 | 0.000491 | 0.022041 |
| 2.5 | 2 | −27,790.90 | 36,206.80 | 35,214.86 | 29,037.57 | 1.212734 | 0.028054 | 0.023133 |
| 2.5 | 4 | −28,018.56 | 79,718.55 | 77,569.88 | 33,511.20 | 2.314745 | 0.061795 | 0.026696 |
| 2.5 | 5 | −28,189.40 | 102,092.00 | 99,246.64 | 36,980.03 | 2.68379 | 0.079064 | 0.02946 |
| 2.5 | 6 | −28,397.47 | 126,183.25 | 122,523.66 | 41,431.65 | 2.957248 | 0.097607 | 0.033006 |
| 2.5 | 7 | −28,638.75 | 151,036.89 | 146,420.90 | 46,832.05 | 3.126511 | 0.116645 | 0.037308 |
| 2.5 | 8 | −28,910.50 | 177,729.00 | 171,975.79 | 53,364.24 | 3.222678 | 0.137003 | 0.042512 |
| 2.5 | 10 | −29,575.00 | 231,178.00 | 222,530.24 | 69,269.33 | 3.212536 | 0.177277 | 0.055183 |

**Table A2.** The force characteristics of the vehicle with the wings deployed under different incoming flow conditions.

| Mach | AOA/° | $F_x$/N | $F_y$/N | Lift Force/N | Drag Force/N | L/D | $C_L$ | $C_D$ |
|---|---|---|---|---|---|---|---|---|
| 8.5 | 0 | −6127.12 | 2122.37 | 2122.37 | 6127.12 | 0.346389 | 0.002709 | 0.007821 |
| 8.5 | 2 | −6357.37 | 24,527.4 | 24,290.59 | 7209.491 | 3.369252 | 0.031005 | 0.009202 |
| 8.5 | 4 | −6854.7 | 47,826.71 | 47,232.05 | 10,174.22 | 4.642324 | 0.060288 | 0.012987 |
| 8.5 | 5 | −7203.52 | 59,811.72 | 58,956.29 | 12,389.04 | 4.758744 | 0.075253 | 0.015814 |
| 8.5 | 6 | −7619.11 | 72,020.3 | 70,829.35 | 15,105.54 | 4.688964 | 0.090408 | 0.019281 |
| 8.5 | 7 | −8101.47 | 84,452.45 | 82,835.63 | 18,333.25 | 4.518328 | 0.105733 | 0.023401 |
| 8.5 | 8 | −8650.6 | 97,108.17 | 94,959.19 | 22,081.26 | 4.300443 | 0.121207 | 0.028185 |
| 8.5 | 10 | −9949.17 | 123,090.3 | 119,492.6 | 31,172.43 | 3.83328 | 0.152522 | 0.039789 |

**Table A2.** *Cont.*

| Mach | AOA/° | $F_x$/N | $F_y$/N | Lift Force/N | Drag Force/N | L/D | $C_L$ | $C_D$ |
|---|---|---|---|---|---|---|---|---|
| 7 | 0 | −10,104.9 | 3115.41 | 3115.41 | 10,104.9 | 0.308307 | 0.002703 | 0.008768 |
| 7 | 2 | −10,431 | 40,526.8 | 40,138.08 | 11,839.01 | 3.390323 | 0.034829 | 0.010273 |
| 7 | 4 | −11,080 | 79,399 | 78,432.69 | 16,591.6 | 4.727252 | 0.068059 | 0.014397 |
| 7 | 5 | −11,510.2 | 99,706.7 | 98,324.11 | 20,156.41 | 4.878056 | 0.08532 | 0.017491 |
| 7 | 6 | −12,004.8 | 120,762 | 118,845.6 | 24,562.1 | 4.838576 | 0.103127 | 0.021314 |
| 7 | 7 | −12,572.22 | 142,549.5 | 139,954.8 | 29,850.93 | 4.688458 | 0.121444 | 0.025903 |
| 7 | 8 | −13,215.62 | 164,733 | 161,290.6 | 36,013.41 | 4.478626 | 0.139958 | 0.03125 |
| 7 | 10 | −14,720.04 | 210,813.6 | 205,054.8 | 51,103.81 | 4.012515 | 0.177934 | 0.044345 |
| 5.5 | 0 | −16,663.7 | 4554.85 | 4554.85 | 16,663.7 | 0.27334 | 0.002923 | 0.010693 |
| 5.5 | 2 | −17,059.4 | 65,103.4 | 64,468.38 | 19,321.08 | 3.336685 | 0.041368 | 0.012398 |
| 5.5 | 4 | −17,848.66 | 127,065.4 | 125,510.8 | 26,668.82 | 4.706277 | 0.080538 | 0.017113 |
| 5.5 | 5 | −18,358 | 159,793 | 157,584.9 | 32,215.02 | 4.89166 | 0.10112 | 0.020672 |
| 5.5 | 6 | −18,953.39 | 192,984.8 | 189,946.4 | 39,021.97 | 4.867679 | 0.121885 | 0.02504 |
| 5.5 | 7 | −19,632.22 | 227,165.4 | 223,079.6 | 47,170.38 | 4.72923 | 0.143146 | 0.030268 |
| 5.5 | 8 | −20,403.2 | 261,765 | 256,377.9 | 56,635.28 | 4.526824 | 0.164513 | 0.036342 |
| 5.5 | 10 | −22,163 | 335,045 | 326,106.3 | 80,006.25 | 4.076011 | 0.209257 | 0.051339 |
| 4 | 0 | −26,719.5 | 6428.71 | 6428.71 | 26,719.5 | 0.2406 | 0.003565 | 0.014815 |
| 4 | 2 | −27,113.8 | 96,994.5 | 95,989.16 | 30,482.34 | 3.149009 | 0.053224 | 0.016902 |
| 4 | 4 | −27,851.4 | 189,619.9 | 187,215.2 | 41,010.77 | 4.565025 | 0.103807 | 0.02274 |
| 4 | 5 | −28,312.9 | 236,749 | 233,380.5 | 48,839.2 | 4.778549 | 0.129404 | 0.02708 |
| 4 | 6 | −28,896.79 | 283,795.3 | 279,220.1 | 58,403.18 | 4.780906 | 0.154821 | 0.032383 |
| 4 | 7 | −29,538.23 | 331,522.4 | 325,451.4 | 69,720.47 | 4.667946 | 0.180456 | 0.038658 |
| 4 | 8 | −30,278.3 | 379,530 | 371,622.5 | 82,804 | 4.487978 | 0.206056 | 0.045913 |
| 4 | 10 | −31,918 | 477,415 | 464,619.5 | 114,335.3 | 4.063656 | 0.257621 | 0.063396 |
| 2.5 | 0 | −33,707.3 | 5621.76 | 5621.76 | 33,707.3 | 0.166782 | 0.004479 | 0.026853 |
| 2.5 | 2 | −33,794.5 | 94,627.4 | 93,390.34 | 37,076.36 | 2.518865 | 0.074399 | 0.029537 |
| 2.5 | 4 | −33,894.62 | 185,676 | 182,859.4 | 46,764.16 | 3.910246 | 0.145673 | 0.037254 |
| 2.5 | 5 | −34,028.7 | 231,213 | 227,367.4 | 54,050.75 | 4.206553 | 0.18113 | 0.043059 |
| 2.5 | 6 | −34,251.26 | 276,781.5 | 271,685 | 62,995.17 | 4.312791 | 0.216436 | 0.050185 |
| 2.5 | 7 | −34,499.9 | 322,564.6 | 315,955.8 | 73,553.48 | 4.295592 | 0.251704 | 0.058596 |
| 2.5 | 8 | −34,785.2 | 368,649 | 360,220.2 | 85,752.7 | 4.200686 | 0.286966 | 0.068314 |
| 2.5 | 10 | −35,543 | 460,633 | 447,463 | 11,4991.1 | 3.891283 | 0.356468 | 0.091607 |

## Appendix B. Position Information of the Iso-Mach Lines in Figures 12–14

In Section 3.2.2, to characterize the Mach flow field around the vehicle, we established corresponding iso-Mach lines and six cross-sections, as illustrated in Figures 12, 13 and 14c,d. To explore the differences among the iso-Mach lines depicted, we measured the intersections of each iso-Mach line with the six cross-sections, establishing eight quantified parameters ($D_{ar}$, $D_{ad}$, $D_{br}$, $D_{bd}$, $D_{isr}$, $D_{lr}$, $D_{isd}$, $D_{ld}$) to detail their distances, resulting in Figures 12, 13 and 14e–j. The values corresponding to these six subfigures are listed in the Tables A3–A5 below.

**Table A3.** The position information of the intersection points of the iso-Mach line to the six cross-sections at Mach 2.5 (Figure 12).

| Name/Unit | Section Number | AOA 0° | AOA 2° | AOA 5° | AOA 7° | AOA 10° |
|---|---|---|---|---|---|---|
| | 1 | 0.156 | 0.162 | 0.165 | 0.174 | 0.180 |
| | 2 | 0.564 | 0.540 | 0.354 | 0.315 | 0.336 |
| $D_{ar}$/m | 3 | 0.894 | 0.726 | 0.324 | 0.336 | 0.408 |
| | 4 | 1.164 | 0.489 | 0.348 | 0.393 | 0.507 |
| | 5 | 1.404 | 0.435 | 0.366 | 0.462 | 0.627 |
| | 6 | 1.626 | 0.345 | 0.366 | 0.504 | 0.735 |

**Table A3.** *Cont.*

| Name/Unit | Section Number | AOA 0° | AOA 2° | AOA 5° | AOA 7° | AOA 10° |
|---|---|---|---|---|---|---|
| $D_{ad}$/m | 1 | 0.150 | 0.156 | 0.162 | 0.174 | 0.186 |
| | 2 | 0.633 | 0.606 | 0.510 | 0.447 | 0.420 |
| | 3 | 0.969 | 0.909 | 0.564 | 0.564 | 0.594 |
| | 4 | 1.254 | 1.065 | 0.630 | 0.639 | 0.744 |
| | 5 | 1.495 | 0.804 | 0.651 | 0.735 | 0.900 |
| | 6 | 1.701 | 0.690 | 0.699 | 0.795 | 1.026 |
| $D_{br}$/m | 1 | −0.153 | −0.153 | −0.153 | −0.150 | −0.150 |
| | 2 | −0.630 | −0.648 | −0.663 | −0.660 | −0.693 |
| | 3 | −0.984 | −0.999 | −1.011 | −1.014 | −1.026 |
| | 4 | −1.266 | −1.281 | −1.290 | −1.269 | −1.284 |
| | 5 | −1.548 | −1.560 | −1.530 | −1.536 | −1.524 |
| | 6 | −1.806 | −1.830 | −1.806 | −1.788 | −1.710 |
| $D_{bd}$/m | 1 | −0.156 | −0.156 | −0.153 | −0.153 | −0.153 |
| | 2 | −0.672 | −0.708 | −0.714 | −0.738 | −0.744 |
| | 3 | −1.020 | −1.056 | −1.077 | −1.074 | −1.095 |
| | 4 | −1.431 | −1.539 | −1.584 | −1.611 | −1.629 |
| | 5 | −1.686 | −1.794 | −1.848 | −1.872 | −1.881 |
| | 6 | −1.920 | −2.058 | −2.115 | −2.130 | −2.130 |
| $D_{isr}/D_{lr}$ ratio | 1 | 1.342 | 1.421 | 1.447 | 1.474 | 1.526 |
| | 2 | 1.517 | 1.569 | 1.586 | 1.629 | 1.681 |
| | 3 | 1.764 | 1.793 | 1.914 | 1.950 | 2.036 |
| | 4 | 2.013 | 2.084 | 2.245 | 2.272 | 2.403 |
| | 5 | 2.294 | 2.362 | 2.519 | 2.619 | 2.756 |
| | 6 | 2.534 | 2.656 | 2.877 | 2.982 | 3.129 |
| $D_{isd}/D_{ld}$ ratio | 1 | 1.281 | 1.281 | 1.282 | 1.282 | 1.284 |
| | 2 | 1.339 | 1.357 | 1.366 | 1.375 | 1.393 |
| | 3 | 1.326 | 1.343 | 1.384 | 1.407 | 1.459 |
| | 4 | 1.388 | 1.405 | 1.445 | 1.467 | 1.489 |
| | 5 | 1.381 | 1.419 | 1.474 | 1.526 | 1.585 |
| | 6 | 1.429 | 1.449 | 1.516 | 1.554 | 1.660 |

**Table A4.** The position information of the intersection points of the iso-Mach line to the six cross-sections at Mach 5.5 (Figure 13).

| Name/Unit | Section Number | AOA 0° | AOA 2° | AOA 5° | AOA 7° | AOA 10° |
|---|---|---|---|---|---|---|
| $D_{ar}$/m | 1 | 0.105 | 0.105 | 0.123 | 0.138 | 0.135 |
| | 2 | 0.375 | 0.354 | 0.330 | 0.309 | 0.309 |
| | 3 | 0.576 | 0.534 | 0.270 | 0.270 | 0.285 |
| | 4 | 0.780 | 0.654 | 0.288 | 0.354 | 0.368 |
| | 5 | 0.870 | 0.339 | 0.330 | 0.423 | 0.435 |
| | 6 | 0.999 | 0.339 | 0.348 | 0.483 | 0.498 |
| $D_{ad}$/m | 1 | 0.108 | 0.108 | 0.108 | 0.114 | 0.126 |
| | 2 | 0.378 | 0.360 | 0.321 | 0.324 | 0.327 |
| | 3 | 0.561 | 0.537 | 0.342 | 0.352 | 0.361 |
| | 4 | 0.738 | 0.696 | 0.354 | 0.363 | 0.375 |
| | 5 | 0.897 | 0.411 | 0.363 | 0.375 | 0.386 |
| | 6 | 1.014 | 0.396 | 0.372 | 0.402 | 0.422 |
| $D_{br}$/m | 1 | −0.105 | −0.105 | −0.105 | −0.096 | −0.084 |
| | 2 | −0.360 | −0.363 | −0.363 | −0.378 | −0.372 |
| | 3 | −0.555 | −0.573 | −0.561 | −0.564 | −0.540 |
| | 4 | −0.726 | −0.783 | −0.786 | −0.789 | −0.789 |
| | 5 | −0.969 | −0.981 | −0.981 | −0.981 | −1.017 |
| | 6 | −1.110 | −1.101 | −1.101 | −1.086 | −1.125 |

**Table A4.** *Cont.*

| Name/Unit | Section Number | AOA 0° | AOA 2° | AOA 5° | AOA 7° | AOA 10° |
|---|---|---|---|---|---|---|
| $D_{bd}$/m | 1 | −0.099 | −0.099 | −0.090 | −0.084 | −0.084 |
| | 2 | −0.366 | −0.375 | −0.354 | −0.363 | −0.366 |
| | 3 | −0.585 | −0.576 | −0.558 | −0.558 | −0.546 |
| | 4 | −0.816 | −0.810 | −0.810 | −0.810 | −0.804 |
| | 5 | −1.026 | −1.044 | −1.035 | −1.050 | −1.020 |
| | 6 | −1.134 | −1.161 | −1.155 | −1.170 | −1.140 |
| $D_{isr}/D_{lr}$ ratio | 1 | 1.332 | 1.335 | 1.343 | 1.349 | 1.356 |
| | 2 | 1.221 | 1.221 | 1.261 | 1.285 | 1.310 |
| | 3 | 1.307 | 1.327 | 1.360 | 1.399 | 1.445 |
| | 4 | 1.420 | 1.450 | 1.540 | 1.599 | 1.659 |
| | 5 | 1.567 | 1.612 | 1.771 | 1.816 | 1.885 |
| | 6 | 1.689 | 1.778 | 1.956 | 2.028 | 2.122 |
| $D_{isd}/D_{ld}$ ratio | 1 | 1.015 | 1.019 | 1.026 | 1.027 | 1.035 |
| | 2 | 1.011 | 1.018 | 1.025 | 1.026 | 1.033 |
| | 3 | 1.012 | 1.017 | 1.026 | 1.029 | 1.033 |
| | 4 | 1.011 | 1.020 | 1.022 | 1.030 | 1.035 |
| | 5 | 1.014 | 1.017 | 1.026 | 1.030 | 1.034 |
| | 6 | 1.013 | 1.017 | 1.026 | 1.027 | 1.032 |

**Table A5.** The position information of the intersection points of the iso-Mach line to the six cross-sections at Mach 8.5 (Figure 14).

| Name/Unit | Section Number | AOA 0° | AOA 2° | AOA 5° | AOA 7° | AOA 10° |
|---|---|---|---|---|---|---|
| $D_{ar}$/m | 1 | 0.091 | 0.099 | 0.119 | 0.099 | 0.110 |
| | 2 | 0.303 | 0.289 | 0.317 | 0.269 | 0.297 |
| | 3 | 0.442 | 0.382 | 0.255 | 0.359 | 0.388 |
| | 4 | 0.552 | 0.396 | 0.286 | 0.376 | 0.427 |
| | 5 | 0.696 | 0.416 | 0.320 | 0.422 | 0.475 |
| | 6 | 0.801 | 0.422 | 0.340 | 0.427 | 0.501 |
| $D_{ad}$/m | 1 | 0.085 | 0.085 | 0.085 | 0.085 | 0.088 |
| | 2 | 0.283 | 0.283 | 0.212 | 0.226 | 0.201 |
| | 3 | 0.425 | 0.382 | 0.260 | 0.255 | 0.255 |
| | 4 | 0.552 | 0.439 | 0.289 | 0.283 | 0.277 |
| | 5 | 0.679 | 0.450 | 0.325 | 0.325 | 0.311 |
| | 6 | 0.778 | 0.416 | 0.342 | 0.337 | 0.325 |
| $D_{br}$/m | 1 | −0.091 | −0.091 | −0.091 | −0.085 | −0.079 |
| | 2 | −0.308 | −0.308 | −0.320 | −0.311 | −0.317 |
| | 3 | −0.467 | −0.467 | −0.470 | −0.456 | −0.461 |
| | 4 | −0.640 | −0.625 | −0.640 | −0.637 | −0.631 |
| | 5 | −0.781 | −0.798 | −0.810 | −0.790 | −0.798 |
| | 6 | −0.877 | −0.908 | −0.912 | −0.892 | −0.892 |
| $D_{bd}$/m | 1 | −0.079 | −0.079 | −0.079 | −0.079 | −0.079 |
| | 2 | −0.280 | −0.297 | −0.311 | −0.311 | −0.317 |
| | 3 | −0.461 | −0.447 | −0.481 | −0.481 | −0.495 |
| | 4 | −0.614 | −0.614 | −0.696 | −0.702 | −0.722 |
| | 5 | −0.784 | −0.792 | −0.877 | −0.877 | −0.900 |
| | 6 | −0.877 | −0.900 | −0.985 | −0.985 | −0.996 |
| $D_{isr}/D_{lr}$ ratio | 1 | 1.272 | 1.274 | 1.277 | 1.280 | 1.282 |
| | 2 | 1.146 | 1.148 | 1.177 | 1.238 | 1.300 |
| | 3 | 1.201 | 1.203 | 1.258 | 1.308 | 1.396 |
| | 4 | 1.263 | 1.286 | 1.377 | 1.463 | 1.571 |
| | 5 | 1.321 | 1.397 | 1.543 | 1.658 | 1.766 |
| | 6 | 1.426 | 1.532 | 1.691 | 1.809 | 1.973 |

| Name/Unit | Section Number | AOA 0° | AOA 2° | AOA 5° | AOA 7° | AOA 10° |
|---|---|---|---|---|---|---|
| $D_{isd}/D_{ld}$ ratio | 1 | 1.011 | 1.014 | 1.016 | 1.021 | 1.026 |
| | 2 | 1.011 | 1.014 | 1.016 | 1.022 | 1.026 |
| | 3 | 1.011 | 1.014 | 1.017 | 1.022 | 1.026 |
| | 4 | 1.012 | 1.015 | 1.017 | 1.022 | 1.027 |
| | 5 | 1.012 | 1.015 | 1.017 | 1.022 | 1.027 |
| | 6 | 1.012 | 1.015 | 1.018 | 1.023 | 1.027 |

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
