# Peer review of "Parametric Design Method and Lift/Drag Characteristics Analysis for a Wide-Range, Wing-Morphing Glide Vehicle"

_aerospace, doi:10.3390/aerospace11040257_

Round 1

Reviewer 1 Report

Comments and Suggestions for Authors

See attached file

It is not clear the link between the results of the aerodynamic analysis (by means of CFD computations) and the design/drawing process described in the UG interface.

I would expect a strong interaction that could lead to an optimized aerodynamic shape. Feedback from CFD results.

Author Response

Thanks for your professional comments. The response is presented in the attachment.

Reviewer 2 Report

Comments and Suggestions for Authors

This paper proposed a new parametric design method for hypersonic vehicles with wing-morphing capabilities. Subsequently, the authors present the lift, drag characteristics, and pressure ratios of these vehicles across a broad operational range. The methodology and insights gained from this work hold some value for the design and optimization of hypersonic vehicles. However, to strengthen the manuscript for publication consideration, I would recommend some revisions. Detailed comments for the authors’ consideration during the revision process are provided below:

1.     On line 80, the authors mentioned "Lobbia investigated various optimization techniques." without providing citations. Similarly, Additionally, in lines 84 and line 86, references are made to [14] and  [15] without specifying the first authors name. Including the first author’s name in these references would increase the clarity and completeness of these sentences.

2.     Figures 12, 13, and 14 each comprise four sub figures. To enhance clarity, the captions for these figures would benefit from explicitly mentioning what each subfigures represents. Briefly describing the content of each subfigure will improve reader comprehension of the overall figures.

3.     In line 404, the authors state “ a larger iso contour when the wings was deployed.” However, the size of these iso contours is not clearly distinguishable. To enhance clarity, it is recommended that the authors provide quantitative values for these parameters. Additionally, creating a grid on y-axis with iso-Mach line values would increase the comprehensibility of the plot.

4.     Line 436, the authors state that “At Mach 5.5, the shock angle decreases.” To strengthen this observation, it would be helpful to quantify the change in shock angle compared to a reference Mach number, such as 2.5 (assuming this is the comparison point). Specifying the actual decrease in angle would provide a clearer understanding of the effect of Mach number on shock behavior. Additionally, clarification of the significance of the red dotted line in figures 12, 13, and 14 is needed for better interpretations.  

5.     In section 3.2.3, the pressure ratio analysis focuses on a configuration with one wing deployed and the other retracted. It would be beneficial to understand the rationale behind choosing this specific configuration for the analysis. Are there specific advantages to investigating pressure ratios in this asymmetric state? Perhaps a brief explanation from the authors would clarify the chosen approach.

6.     There seems to be a discrepancy in the discussion of Figure 16. The authors state, “As the inflow Mach number increases, the pressure ratio strengthens”. However, figure 16 depicts variations in angle of attack is changing while keeping the Mach number fixed at 5.5. Is appears that the authors might be referring to figure 17 instead of figure 16. Clarification on this point is needed for better coherence in the manuscript.

Minor comments:

Line 456: “Mach number distribution in the flow field” to “effect of Mach number variations in the flow field”.

Line 53: “are also play” to “also plays”

Comments on the Quality of English Language

Moderate editing of the English is needed in the manuscript.  

Author Response

(The authors gave the same response as above.)

Round 2

Reviewer 1 Report

Comments and Suggestions for Authors

See attached file

Author Response

Thanks for your professional comments. 

We are writing directly based on Aerospace's templates, and there should be no problem with font errors. However, after we saved the manuscript as PDF, there were indeed some 'Cross-reference' errors, which might be the font problem you pointed out.
Please see the attachment for details.

Reviewer 2 Report

Comments and Suggestions for Authors

The authors have effectively addressed all of my inquiries and concerns. I will recommend it for publication. However, I have noticed the presence of Chinese characters in the revised manuscript, particularly on pages 6-12. These characters must be replaced with English text before finalizing the publication. 

Comments on the Quality of English Language

Minor editing of the English language is required. 

Author Response

(The authors gave the same response as above.)
